# FOXP3+ regulatory T cell perturbation mediated by the IFNγ-STAT1-IFITM3 feedback loop is essential for anti-tumor immunity

Xinnan Liu[1,16], Weiqi Zhang[1,16], Yichao Han[2,16], Hao Cheng[3,16], Qi Liu[4,16], Shouyu Ke[5], Fangming Zhu [6], Ying Lu[7], Xin Dai[8,9], Chuan Wang[10], Gonghua Huang [1,11], Bing Su [1], Qiang Zou[1], Huabing Li [1], Wenyi Zhao[5], Lianbo Xiao[9], Linrong Lu[12], Xuemei Tong [4], Fan Pan [3,17] ✉, Hecheng Li [2,17] ✉ & Bin Li [1,2,9,13,14,15,17] ✉

Targeting tumor-infiltrating regulatory T cells (Tregs) is an efficient way to evoke an anti-tumor immune response. However, how Tregs maintain their fragility and stability remains largely unknown. IFITM3 and STAT1 are interferon-induced genes that play a positive role in the progression of tumors. Here, we showed that IFITM3-deficient Tregs blunted tumor growth by strengthening the tumor-killing response and displayed the Th1-like Treg phenotype with higher secretion of IFNγ. Mechanistically, depletion of IFITM3 enhances the translation and phosphorylation of STAT1. On the contrary, the decreased IFITM3 expression in STAT1-deficient Tregs indicates that STAT1 conversely regulates the expression of IFITM3 to form a feedback loop. Blocking the inflammatory cytokine IFNγ or directly depleting STAT1-IFITM3 axis phenocopies the restored suppressive function of tumor-infiltrating Tregs in the tumor model. Overall, our study demonstrates that the perturbation of tumor-infiltrating Tregs through the IFNγ-IFITM3-STAT1 feedback loop is essential for anti-tumor immunity and constitutes a targetable vulnerability of cancer immunotherapy.

FOXP3 expressing regulatory T cells (Tregs) can suppress autoimmune response but also affect anti-tumor immunity[1]. Tregs are found throughout the tumor microenvironment (TME) and can exert suppressive function considerably, forming physical, metabolic, and trafficking 'barriers' to exclude pro-inflammatory cells from the TME[2]. In humans, depleting Treg cells to achieve anti-tumor immunity is under clinical trial, and expanding the population of Treg cells to treat autoimmune diseases has also shown considerable efficacy[3]. In the TME, altering Treg function by inducing Treg fragility, with lower expression of FOXP3, enhanced production of tumor-killing cytokines such as interferon-gamma (IFNγ), could contribute to the anti-tumor

therapy[4]. However, it is still challenging to understand how Treg cell fragility and stability are regulated in the TME and whether this regulation can be helpful in fine-tuneing the immune responses.

As a member of type II interferon, IFNγ became well known for its immune-modulating effects on innate and adaptive immunity[5]. IFNγ could induce apoptosis of tumor cells through the caspase pathway and Fas/FasL-induced cell death. It can regulate the presentation of antigens, promotion of inflammation, and activation and polarization of responding leukocytes. Besides, it also exerts direct antiproliferative effects on T cells[6]. Interestingly, IFNγ has been classically considered a pro-inflammatory cytokine, which negatively regulates anti-

inflammatory responses by antagonizing the IL10 and TGF-β signaling pathways to suppress the Treg cell function[7]. On the other hand, the increased IFNγ secretion from T-helper-1 (Th1)-like Treg cells is often accompanied by impaired Treg immunosuppressive function and Treg lineage stability[8]. From our previous data, E3 deubiquitinase USP21 prevents the depletion of FOXP3 and restricts the generation of Th1-like Treg cells[9]. Strikingly, the differentiation of Treg cells towards a Th1-like phenotype can contribute to cancer immune therapy[10].

Interferon-induced transmembrane 3 (IFITM3) is a small antiviral effector protein induced by interferon cytokines to inhibit cellular infection of influenza A and many other viruses[11]. The previous research in our lab has identified lysine methyltransferase SET7 and histone demethylase LSD1 to regulate the methylation of IFITM3, thus modulating the antiviral function of IFITM3[12,13]. Except for innate immune response, IFITM3 has been found to strengthen resident memory T cells ($T_{RM}$) function with greater resistance to viruses[14]. Moreover, IFITM3 is also overexpressed in multiple types of cancer cells, and its expression correlates with histopathological grading and staging[15]. IFITM3 has been reported to function as an amplifier of PI3K signaling at the cell surface of B cells, which is critical for the rapid expansion of B cells with high affinity to antigen[16]. But whether and how IFITM3 acts to control Treg cell functions in anti-tumor immunity remains largely unexplored.

Here, we identified a negative feedback loop in which reduced IFITM3 expression induces higher phosphorylation of STAT1, which promotes IFNγ responses in tumor-infiltrating Treg cells (TI-Tregs) and induces the fragility of TI-Tregs. We validated this loop using mouse tumor models and murine cells. In TI-Tregs, the feedback loop of STAT1-IFITM3 was associated with Treg function, and the perturbation of this loop may affect anti-tumor immunity. Disrupting this circuit genetically, either by knocking out IFITM3 or STAT1, has therapeutic efficacy in tumor models by abolishing the function of TI-Tregs. Altogether, this study provides insights for clinical anti-tumor therapy and suggests that IFITM3 may emerge as an immune checkpoint on Treg cells.

## Results

### Tumor-infiltrating Tregs are characterized by a positive correlation between IFITM3 and FOXP3

To investigate the potential function of IFITM3 in the tumor microenvironment, we first explored the TCGA database to analyze the mRNA expression profile of IFITMs in different cancer types. We found that among various IFITMs, only IFITM1 and IFITM3 were highly expressed in tumor tissues from both colon adenocarcinoma (COAD) and esophageal carcinoma (ESCA) patients as compared to normal tissue (Fig. 1a and Supplementary Fig. 1a, b). However, validation of these findings in tissues from COAD patients using the RT−qPCR showed that *IFITM3*, but not *IFITM1*, was highly expressed in the tumor (Fig. 1b and Supplementary Fig. 1c), which indicated the positive association of IFITM3 expression with the disease progression. To investigate the expression profile of IFITM3 in TI-Tregs, we next detected the IFITM3 expression in Treg cells isolated tumor tissue and corresponding normal tissue of COAD ($n = 4$) and ESCA ($n = 6$) patients. The results showed that the expression of IFITM3 was upregulated in Treg cells from both ESCA and COAD tissue (Fig. 1c and Supplementary Fig. 1d). Similar to the reported results that the tumor microenvironment is capable of recruiting more Treg cells[17], we found that there was a higher proportion of Treg cells in ESCA tumor tissue compared to normal tissue (Fig. 1d). Based on the expression of FOXP3, TI-Tregs could be divided into two groups: FOXP3[low], and FOXP3[high] (Fig. 1d, gating strategy is showed in Supplementary Fig. 1e). We found that FOXP3[high] Treg subsets showed higher expression of IFITM3 (Fig. 1e and Supplementary Fig. 1f). In addition, the correlation analysis of IFITM3 expression and FOXP3 expression of TI-Tregs of

ESCA patients also demonstrated that IFITM3 may participate in the regulation of Treg cell function and stability in the TME (Fig. 1f). Subsequently, we verified these findings using MC38 syngeneic mouse tumor model. Similarly to our clinical data, Ifitm3 mRNA level was greatly increased in Treg cells from MC38 tumor tissue (Fig. 1g). Collectively, these results indicate that IFITM3 is associated with tumor progression and may exert substantial roles in the regulation of TI-Treg cells.

### Depletion of IFITM3 in Tregs does not influence T-cell development

To determine the function of IFITM3 in Treg cells, we generated mice that conditionally deleted *Ifitm3* in Treg cells (*Ifitm3[fl/fl]Foxp3[YFP-cre]*) (Supplementary Fig. 2a). RT−qPCR analysis and immunoblot data showed that the expression of IFITM3 was diminished at mRNA and protein level, respectively, in peripheral Treg cells from *Ifitm3[fl/fl]Foxp3[YFP-cre]* mice (Supplementary Fig. 2b, c). Since Treg cells develop from the thymus, we first examined the thymic T-cell development. We found that thymocyte development was normal in *Foxp3[YFP-cre]* (WT) and *Ifitm3[fl/fl]Foxp3[YFPcre]* (cKO) mice, as indicated by the percentage and cell number of thymocyte subsets: CD4[−]CD8[−] double negative (DN), CD4[+]CD8[+] double positive (DP), and CD4[+]/CD8[+] single positive cells (Supplementary Fig. 2d). Furthermore, in the peripheral lymph nodes (pLNs) and spleen, there was also no difference in the percentage and number of CD4[+] and CD8[+] T cells between WT and cKO mice (Supplementary Fig. 2e). In addition, the percentage of FOXP3[+] Treg cells in CD4[+] T cells and FOXP3 protein level were almost identical between WT and cKO mice (Supplementary Fig. 2f). This indicates that IFITM3 deficiency does not influence the thymocyte development and natural Treg cell development or maintenance. Due to the function of Treg cells in maintaining long-lasting homeostasis, we further detected the activation of CD4[+] and CD8[+] T cells and the secretion of IFNγ from CD4[+] and CD8[+] T cells in young IFITM3 cKO mice (6-8 weeks) and elder IFITM3 cKO mice (12-month). Fascinatingly, we found increased activation of CD4[+] and CD8[+] T cells (Supplementary Fig. 3a, b, e, f) and IFNγ secretion from CD4[+] and CD8[+] T cells (Supplementary Fig. 3c, d, g, h) in elder cKO mice but not in 6-8 weeks old mice, which indicated that IFITM3 might impact Treg stability and function to regulate immune homeostasis.

### IFITM3 deficiency in Tregs promotes anti-tumor responses in vivo by damaging Treg suppressive function

Since IFITM3 was associated with tumor progression and its deficiency-induced instability of Treg cells, we hypothesized that IFITM3 was involved in regulating Treg cells in the TME. To address this hypothesis, we inoculated WT and cKO mice with MC38 murine colon cancer cells subcutaneously to determine whether *Ifitm3* is required for Treg cells in anti-tumor responses in vivo. Compared with WT mice, cKO mice had profound reductions in MC38 tumor size (Fig. 2a, b). The tumor-induced lethality of cKO mice was also weaker than that of WT mice (Fig. 2c). In addition, MC38 tumor cell-challenged cKO mice had increased infiltration of CD8[+] TILs, and NK cells in the TME (Fig. 2d, e). Conventional CD4[+] T (Tconv) cells and CD8[+] TILs showed higher proliferation in cKO mice (Fig. 2f). The production of IFNγ and TNFα, which play a primary role in eliminating tumors, was also increased in Tconv cells and CD8[+] TILs (Fig. 2g, h). Collectively, these results suggest that the absence of IFITM3 in Treg cells contributes to the infiltration of tumor-killing cells and the production of effector cytokines in the TME, leading to a more robust anti-tumor response. Interestingly, cKO mice challenged with MC38 tumor cells displayed a significant increase in the percentage of TI-Tregs (Fig. 3a). However, these TI-Treg cells exhibited decreased FOXP3 expression (Fig. 3b). Further analysis indicated that IFITM3-deficient TI-Treg cells were more activated (higher expression of CD25) and proliferative (higher expression of Ki67)

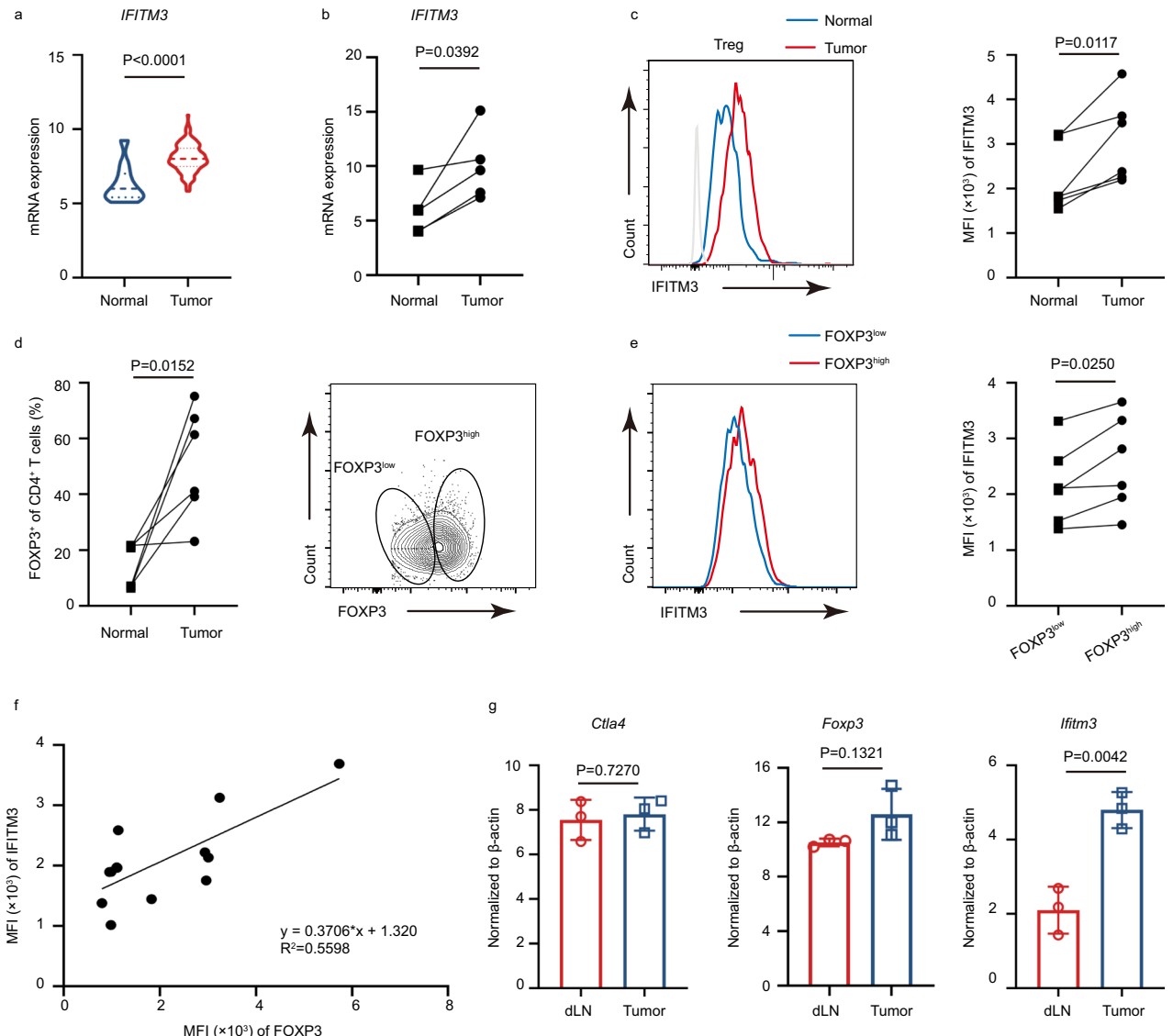

**Fig. 1 | IFITM3 is correlated with Tregs in the tumor microenvironment. a** mRNA expression of *IFITM3* in TCGA ESCA database. **b** qRT-PCR analysis of *IFITM3* in tissue from COAD patients (*n* = 6). **c** Histogram shows the MFI of IFITM3 and the quantification of the MFI of IFITM3 in Treg cells from normal tissue and tumor tissue from ESCA patients (*n* = 6). **d** Treg percentage of CD4⁺ T cells in normal tissue and tumor tissue of ESCA patients. Tumor-infiltrating Treg cells were divided into two subsets (FOXP3ʰⁱᵍʰ and FOXP3ˡᵒʷ)according to FOXP3 expression (*n* = 6).

**e** Histogram shows the MFI of IFITM3 and the quantification of the MFI of IFITM3 in FOXP3ʰⁱᵍʰ and FOXP3ˡᵒʷ tumor Treg cells (*n* = 6). **f** Correlation analysis of IFITM3 expression and FOXP3 expression in tumor-infiltrating Treg cells from ESCA patients (*n* = 13). **g** qRT-PCR analysis of *Ifitm3* and genes associated with Treg function in Treg cells from the tumor and normal tissue (dLN) of MC38 tumor-bearing mice (*n* = 3). Data are represented as the mean ± SD. Source data are provided as a Source Data file.

compared to TI-Tregs in WT mice (Fig. 3c, d). Meanwhile, decreased suppressive cytokine IL10 and increased tumor-killing cytokine IFNγ expression were detected in IFITM3-deficient TI-Treg cells (Fig. 3e, f). These results further support that the immunosuppressive function of TI-Tregs is impaired in the absence of IFITM3. However, there was no significant difference in peripheral Treg cells in the tumor model (Fig. 3g–j), indicating that the role of IFITM3 in Treg cells was dependent on the tumor microenvironment.

**IFITM3-deficient Tregs exhibit higher phosphorylation of STAT1 with increased nuclear translocation**

To clarify the mechanism underlying the reduced suppressive capacity of IFITM3-deficient Tregs, we isolated TI-Tregs from MC38 tumor-bearing WT and cKO mice for transcriptomic analysis. We found that IFITM3-deficient Tregs displayed high expression of interferon-related genes (*Isg15*, *Stat1*, *Tbx21*, *Irf1*, *Ifi44*) and increased expression of Tregs-

activation markers (*Il2ra*, *Akt2*) (Fig. 4a). Furthermore, IFITM3-deficient Tregs exhibited enrichment in IFNγ response-related genes (Fig. 4b). The expression of aberrantly expressed genes was further validated in TI-Tregs by RT-qPCR (Fig. 4c). These data indicate that STAT1, a key mediator of IFNγ signaling that regulates the differentiation of Treg cells[18], was highly increased in IFITM3-deficient Tregs (Fig. 4c). It has been known that STAT1 is phosphorylated and translocated into the nucleus to mediate the transcription of the downstream target genes[19]. Immunoblot, and flow cytometric analyses revealed that, upon TCR stimulation, the expression of STAT1 and the activation of STAT1 (assessed by measuring the phosphorylation of STAT1 at Tyr701) were enhanced in IFITM3-deficient Tregs compared to Tregs from WT mice (Fig. 4d–f). ChIP-qPCR analysis also indicated that IFITM3-deficient Tregs showed higher transcriptional levels of STAT1-regulated genes (*Ifi44* and *Psmb9*) (Fig. 4g).

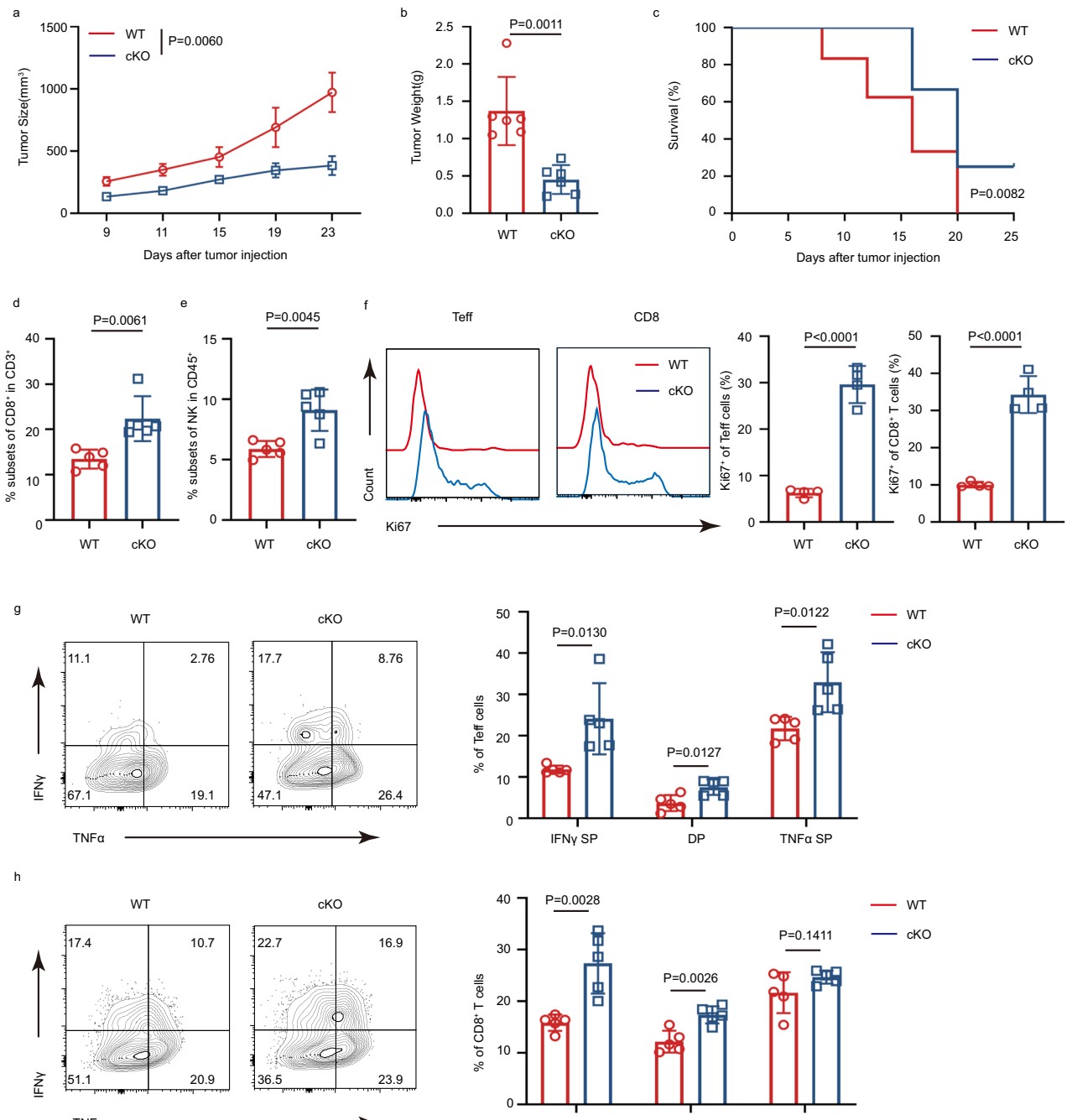

**Fig. 2 | IFITM3 deficient in Tregs promotes anti-tumor responses in vivo.**
**a**, **b** Tumor growth (**a**, WT group: $n = 7$; cKO group: $n = 7$) and tumor weight (**b**, WT group: $n = 6$; cKO group: $n = 6$) in WT and cKO mice injected s.c. with MC38 murine colon cancer cells. **c** Survival curve of WT and cKO mice injected s.c. with MC38 murine colon cancer cells ($n = 7$ per group). **d**, **e** Percentage of CD8+ (**d**) and NK cells (**e**) in tumor of WT and cKO mice injected s.c. with MC38 murine colon cancer cells (day 23, $n = 5$ per group). **f** Histogram shows the MFI of Ki67 and the quantification of the MFI of Ki67 in CD8+ T cells and Teff cells in tumors of WT and cKO mice injected s.c. with MC38 murine colon cancer cells ($n = 4$ per group). **g**, **h** Cytokine secretion of CD8+ and Teff cells in tumor of WT and cKO mice injected s.c. with MC38 murine colon cancer cells (day 23, $n = 5$ per group). Survival curves were analyzed by log-rank (Mantel-Cox) test. The data **d**–**h** are presented as representative plots and as summary graphs ($n = 4/5$ per group). Source data are provided as a Source Data file.

## IFITM3 suppresses JAK- STAT1 activation by binding to JAK-STAT1 protein complex

To further explore the underlying mechanism, we separated nuclear and cytoplasmic proteins to observe whether IFITM3 deficiency could alter the intracellular localization of STAT1, mainly reflected by increased STAT1 translocation in the nucleus of Tregs isolated from cKO mice (Supplementary Fig. 4a). To establish how IFITM3 regulates the expression and nuclear translocation of STAT1, we performed co-immunoprecipitation to determine whether IFITM3 directly interacted with STAT1. We found that IFITM3 could interact with STAT1 in mouse Treg cells, HEK293T cells, and Hela cells (Fig. 5a and Supplementary Fig. 4b, c). Interestingly, when HA-STAT1 and Myc-IFITM3 were transfected into HEK293T cells, HA-STAT1 protein was decreased, accompanied by increased Myc-IFITM3 (Fig. 5b). In addition, the decrease of HA-STAT1 protein mainly occurred in the nucleus, indicating that IFITM3 regulates nuclear translocation of STAT1 (Fig. 5c). The previous studies showed that phosphorylation site mutation Tyr20 (Y20A, Y20E, and ΔY20) could induce the membrane location of IFITM3[20,21]. In

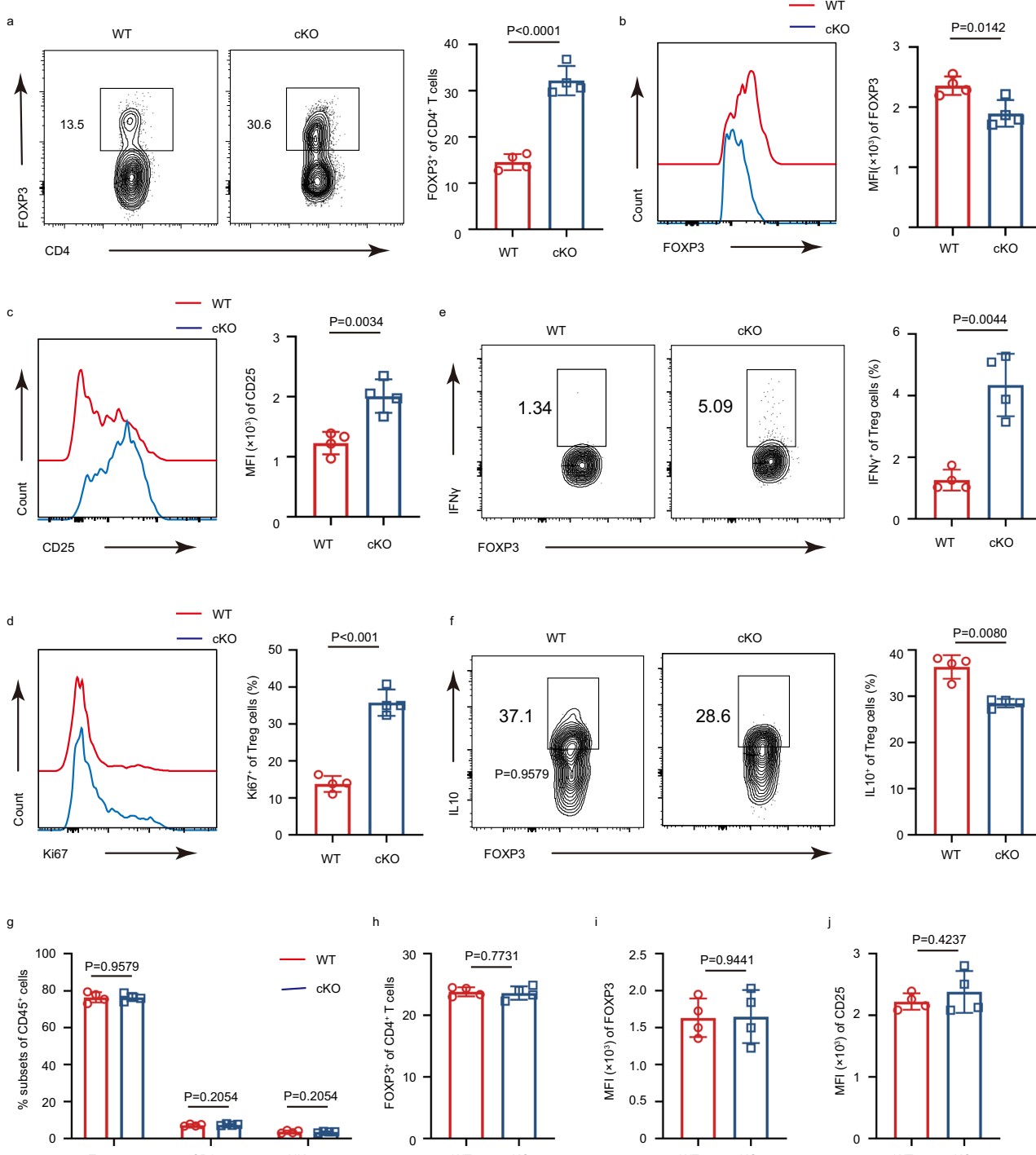

**Fig. 3 | IFITM3 ablation dampens Treg function and stability in the tumor.**
**a** Percentage of CD4+FOXP3+ Treg cells among CD4+ T cells in tumor of WT and cKO mice injected s.c. with MC38 murine colon cancer cells (day 23, *n* = 4 per group). **b** Histogram shows the MFI of FOXP3 and the quantification of the MFI of FOXP3 in tumor Treg cells of WT and cKO mice injected s.c. with MC38 murine colon cancer cells (day 23, *n* = 4 per group). **c** Histogram shows the MFI of CD25 and the quantification of the MFI of CD25 in tumor Treg cells of WT and cKO mice injected s.c. with MC38 murine colon cancer cells (day 23, *n* = 4 per group). **d** Histogram shows the MFI of Ki67 and the quantification of the MFI of Ki67 in tumor Treg cells of WT and cKO mice injected s.c. with MC38 murine colon cancer cells (day 23, *n* = 4 per

group). **e**, **f** Frequency of IFNγ (**e**) and IL10 (**f**) producing Treg cells in tumor of WT and cKO mice injected s.c. with MC38 murine colon cancer cells (day 23, *n* = 4 per group). **g** Percentage of Tconv cells, CD8+ T cells, and NK cells in the spleen of WT and cKO mice injected s.c. with MC38 murine colon cancer cells (day 23, *n* = 4 per group). **h** Percentage of CD4+FOXP3+ Treg cells among CD4+ T cells in the spleen of WT and cKO mice injected s.c. with MC38 murine colon cancer cells (day 23, *n* = 4 per group). **i**, **j** The quantification of the MFI of FOXP3 (**i**) and CD25 (**j**) in spleen Treg cells of WT and cKO mice injected s.c. with MC38 murine colon cancer cells(day 23, *n* = 4 per group). The data are presented as summary graphs (*n* = 4 per group). Data are represented as the mean ± SD. Source data are provided as a Source Data file.

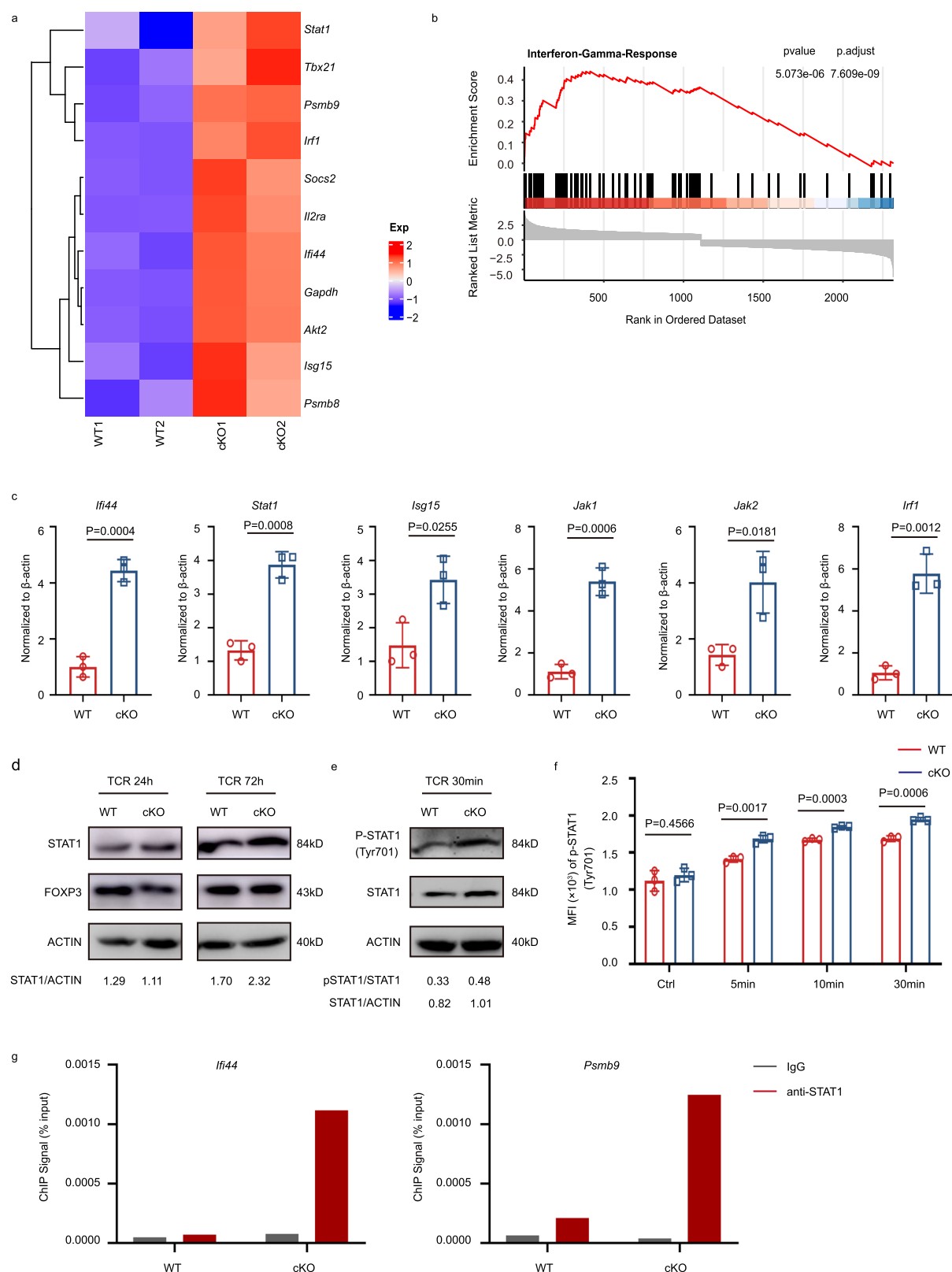

addition, the phosphomimetic IFITM3 (Y20E) mutation also induced constitutive localization of this protein at the plasma membrane in B cell leukemia[16]. We, therefore, speculated whether the cell membrane localization of IFITM3 would enhance its regulation of STAT1 protein levels. Similarly, the transfection of the mutant IFITM3 plasmids and HA-STAT1 into HEK293T cells further decreased the expression of STAT1 (Supplementary Fig. 4d).

Type I and type II cytokine receptors are selectively associated with JAKs, and activation of the receptor-bound JAKs is critical for initiating phosphorylation of the cytokine receptor and subsequent recruitment of one or more STATs[22]. We confirmed that IFITM3 could also interact with JAK1 and JAK2 (Fig. 5d, e and Supplementary Fig. 4e). Moreover, our data indicate that transfection of HEK293T cells with IFITM3 and JAK1/JAK2 elevates the location of IFITM3 on the cell

**Fig. 4 | IFITM3 is dispensable for STAT1 phosphorylation and expression in Tregs. a** Heatmap of upregulated genes associated with Treg activation and IFN signal pathway in tumor-infiltrating WT and IFITM3-deficient Treg cells. Treg cells were isolated from tumor-bearing WT and KO mice on day 23 after the injection of MC38 murine colon cancer cells. **b** GSEA enrichment plots of the indicated signatures in tumor-infiltrating WT and IFITM3-deficient Treg cells. **c** qRT-PCR analysis of upregulated genes in tumor-infiltrating Treg cells of WT and cKO mice (*n* = 3). **d** Immunoblot analysis of STAT1 and FOXP3 in WT and cKO Treg cells stimulated

with anti-CD3 and anti-CD28 antibodies for the indicated durations. **e** Immunoblot analysis of p-STAT1(Tyr701) and STAT1 in WT and cKO Treg cells stimulated with anti-CD3 and anti-CD28 antibodies for the indicated durations. **f** Flow cytometric analysis of p-STAT1(Tyr701) in WT and cKO Treg cells stimulated with anti-CD3 and anti-CD28 antibodies for the indicated durations. **g** ChIP-qPCR analysis of *Ifi44* and *Psmb9* that are regulated by STAT1 in WT and cKO Treg cells. Data in **d–f** are representative of 3 independent experiments. Data are represented as the mean ± SD. Source data are provided as a Source Data file.

membrane (Fig. 5f). Thus, our findings demonstrate that IFITM3 could regulate the phosphorylation and nuclear translocation of STAT1 by binding to the JAK-STAT1 axis on the cell membrane.

Besides STATs, JAK could also phosphorylate PI3K and activate the PI3K-AKT signaling pathway[23]. Upon PI3K activation, AKT phosphorylates FOXO1, causing its exportation from the nucleus into the cytoplasm, which ultimately increases the expression of IFNγ in Treg cells accompanied by FOXP3 instability[24]. From our RNA-seq data, we also observed low expression of Treg cell stability markers (*Icos*, *Klrg1*, *Cd28*, *Il7r*) and AKT-FOXO axis-related genes (Fig. 5g). Since we have defined that IFITM3 could translocate from cytoplasm to cell membrane to participate in the JAK-STAT1 signaling pathway, we hypothesized that IFITM3 may also regulate JAK-PI3K signaling. Indeed, our results also revealed that IFITM3 deficiency inhibits the phosphorylation of AKT at Ser473 and decreases the FOXO1 protein level in Tregs (Fig. 5h, i). Therefore, the upstream pathway of PI3K/FOXO1 signaling is also regulated by IFITM3, resulting in increased secretion of IFNγ and decreased stability of FOXP3.

## IFITM3 interacts with STAT1 and induces its autophagosome-mediated degradation

It has been reported that overexpression of IFITM3 can induce autophagy, including LC3 puncta formation and lipidation[25,26]. It was found that IFITM3 promotes the transformation of type I LC3 to type II LC3, an indicator for enhanced autophagy, in a dose-dependent manner[27]. From our data, IFITM3 could influence the phosphorylation, translocation, and stability of STAT1. Since proteases and autophagy mediate the protein degradation[28], we first confirmed that STAT1 could interact with LC3, indicating the autophagy degradation of STAT1, and IFITM3 could also interact with LC3 (Supplementary Fig. 5a–c). To investigate the role of IFITM3 in STAT1 degradation, we treated HEK293T cells transfected with IFITM3 and STAT1 with proteasome inhibitor MG132 and autophagy inhibitor chloroquine. MG132 treatment had little effect on the degradation of STAT1 caused by expression of IFITM3 (Supplementary Fig. 5d). However, the autophagy inhibitor chloroquine treatment reversed the degradation of STAT1 caused by overexpression of IFITM3, suggesting that IFITM3-mediated degradation of STAT1 was autophagy-dependent but not proteasome-dependent. Additionally, IFITM3 in conjunction with LC3 resulted in progressive STAT1 reduction (Supplementary Fig. 5e). Therefore, IFITM3 could participate in and regulate the autophagic degradation of STAT1.

## STAT1 and IFITM3 form an IFNγ-dependent feedback loop

From our data, IFITM3 suppresses the transcription function and protein stability of STAT1 to maintain Treg function in the tumor microenvironment. Since IFNγ and STAT1 induce IFITM3, an important transcriptional factor under IFNγ stimulation, we speculated whether STAT1 also regulates IFITM3. We found that IL12, IL27, and IFNγ, which could increase the STAT1 expression to induce Th1 response in the tumor microenvironment, could also induce high expression of IFITM3 of Treg cells (Fig. 6a). But this upregulation of IFITM3 was not found in Treg cells from *Stat1*^fl/fl^*Foxp3*^YFP-cre^ mice(SKO) (Fig. 6b, c). In addition, STAT1 could bind to the *Ifitm3* promoter and regulate the transcription of *Ifitm3* (Fig. 6d). These results indicate that STAT1 is an essential factor for IFITM3 expression; thus, IFITM3 and STAT1 form a feedback

loop, and they can regulate each other to maintain the activation of the STAT1 signaling pathway.

IFNγ could induce the expression of IFITM3 and STAT1, and the IFITM3 cKO mice showed higher IFNγ expression in the tumor environment to mediate a more robust anti-tumor response. We then investigated whether IFNγ is essential for the feedback loop of IFITM3 and STAT1. We crossed IFNγ^+/−^ mice with IFITM3 cKO mice (to generate IFNγ^+/−^ cKO mice) and inoculated them with MC38 murine colon cancer cells. Compared with cKO mice, IFNγ^+/−^ cKO mice showed a significant increase in tumor size and weight (Fig. 6e, f). IFNγ^+/−^ cKO mice showed a comparable level of frequency and cell number of NK cells, CD8^+^ T cells, CD4^+^ T cells, and Tregs infiltration into the tumors with WT mice (Fig. 6g, Supplementary Fig. 6a–g). Furthermore, Tregs in IFNγ^+/−^ cKO mice secreted higher levels of IL10 and showed a lower proliferation rate than cKO mice (Fig. 6h–j), indicating that blocking IFNγ in IFITM3 cKO mice could restore the suppressive function of TI-Tregs. As expected, the proliferation rate of CD4^+^ T cells and CD8^+^ T cells in the tumor microenvironment from IFNγ^+/−^ cKO mice were considerably rescued by the deletion of IFNγ (Supplementary Fig. 6h). Whereas, tumor-infiltrating T cells in IFNγ^+/−^ cKO mice almost did not secret IFNγ (Fig. 6k, l). These data support the hypothesis that IFITM3 could form a feedback loop with STAT1 in Treg cells, and IFNγ is essential for the homeostasis of this loop. Strikingly, the anti-IFNγ treatment of the cKO mice tumor model also showed similar tumor growth with WT mice, consistent with the IFNγ KO mice model (Supplementary Fig. 6i–k).

## The perturbation of the TI-Treg function mediated by the IFNγ-STAT1-IFITM3 feedback loop is essential for TI-Treg function and stability

To better understand the importance and function of the STAT1-IFITM3 loop in Treg, we induced MC38 tumors in IFITM3 conditional knock-out mice (cKO), STAT1 conditional knock-out mice (SKO), and IFITM3/STAT1 double conditional knock out mice (DKO). The SATT1 cKO mice and DKO mice showed similar T-cell percentage and Thymus development (Supplementary Fig. 7a–e). Although the activation of CD4^+^ and CD8^+^ T effector cells showed a slight elevation in SKO mice (Supplementary Fig. 7f, g), the IFNγ secreting CD4^+^ and CD8^+^ T effector cells did not show any significant change in both SKO and DKO mice, indicating that the IFITM3-STAT1 feedback loop did not influence the T cells development and immune hemostasis (Supplementary Fig. 7h, i).

Then, we used the MC38 tumor model to evaluate the function of the IFITM3-STAT1 feedback loop in Tregs. Compared with WT mice, both cKO and SKO mice showed decreased tumor growth and weight. However, the tumor growth and tumor weight of DKO mice were similar to WT mice (Fig. 7a, b). Similar changes were also found in the Tregs percentage, the number of Tregs, CD8^+^ T cells, and NK cells (Fig. 7c–f). cKO mice and SKO mice also increased IFNγ secreting CD4^+^ Tconv cells and CD8^+^ T cells; however, the DKO mice exhibited identical IFNγ levels with WT mice (Fig. 7g, h). We further performed ELISA to study the anti-tumor-specific response of IFITM3 cKO, STAT1 cKO, and double KO ex vivo by stimulating splenocytes against the MC38 cell line. The results showed that STAT1-deficient and IFITM3-deficient Tregs increased the IFNγ secretion in the coculture system. However, double-deficient Tregs showed reduced IFNγ levels (Supplementary

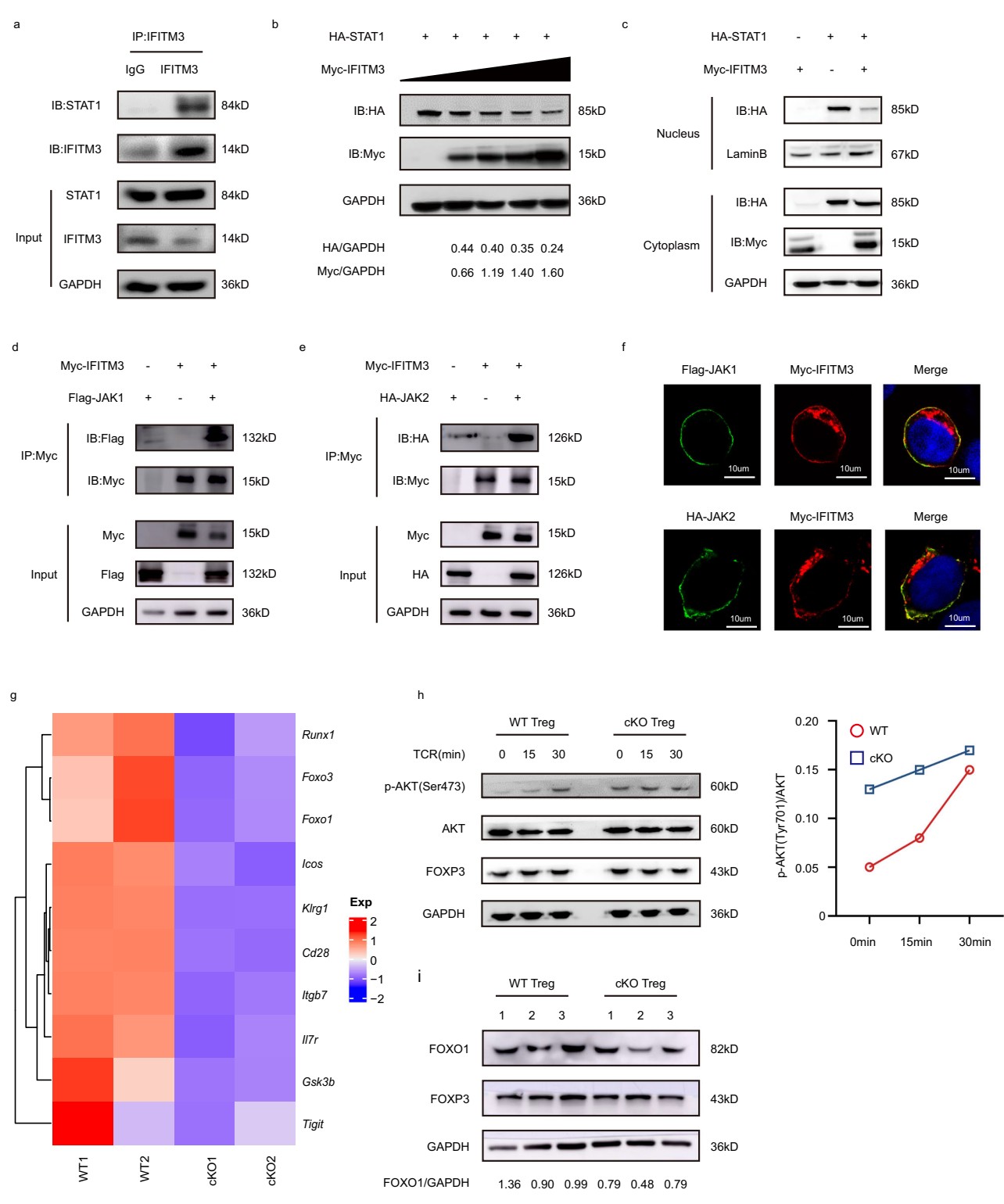

Fig. 8a). The in vivo suppression assay indicated both IFITM3-deficient and STAT1-deficient Tregs suppress Treg function. In contrast, Tregs from double KO restored the Treg suppression function (Supplementary Fig. 8b). Similar changes were found in cytokine production from T effector cells and Treg cells (Supplementary Fig. 8c, d). These results indicated that both IFITM3 deficiency and STAT1 deficiency disrupt the IFITM3-STAT1 feedback loop in TI-Tregs and could enhance the anti-tumor immune response by suppressing the immunosuppressive functions of TI-Tregs. On the contrary, knocking out both IFITM3 and STAT1 in TI-Tregs restores the suppression function and inhibits the anti-tumor response. These results suggest that breaking the balance between STAT1 and IFITM3, rather than completely removing the loop, is beneficial for Treg function and anti-tumor immunity (Fig. 8).

**Fig. 5 | IFITM3 mediates JAK-STAT1 activation and facilitates the AKT-FOXO1 axis. a** Cell lysates of mouse Treg cells were immunoprecipitated with anti-IFITM3 antibody and assessed by immunoblotting with anti-STAT1. **b** HEK293T cells were transfected with HA-tagged STAT1 and increased amounts of Myc-tagged IFITM3. The level of Myc-IFITM3 was then detected by immunoblotting. **c** Immunoblot analysis of HA-STAT1 and Myc-IFITM3 in nuclear and cytoplasm extracted from HEK293T cells transfected with HA-tagged STAT1 and Myc-tagged IFITM3. **d** HEK293T cells were transfected with the Flag-tagged JAK1 and Myc-tagged IFITM3 were immunoprecipitated with anti-Myc antibody and assessed by immunoblotting with anti-Flag. **e** HEK293T cells were transfected with the HA-tagged JAK2 and Myc-tagged IFITM3 were immunoprecipitated with anti-Myc antibody and assessed by immunoblotting with anti-HA. **f** Immunofluorescence analysis of HEK293T cells transfected with Flag-JAK1/HA-JAK2 and Myc-IFITM3. **g** Heatmap of downregulated genes associated with Treg activation and IFN signal pathway in tumor-infiltrating WT and IFITM3-deficient Treg cells. Treg cells were isolated from tumor-bearing WT and KO mice on day 23 after the injection of MC38 murine colon cancer cells. **h** Immunoblot analysis of p-AKT(Ser473) and AKT in WT and cKO Treg cells stimulated with anti-CD3 and anti-CD28 antibodies for the indicated durations. **i** Immunoblot analysis of FOXO1 in WT and cKO Treg cells. Experiments were independently repeated 3 times. Source data are provided as a Source Data file.

## IFITM3 also regulates CD4$^+$ and CD8$^+$ TIL function in the tumor

Besides elucidating the role of IFITM3 in Tregs, we also detected the IFITM3 expression in tumor-infiltrating CD4$^+$ and CD8$^+$ T cells to check its potential functions. We performed RT−qPCR analysis and determined *Ifitm3* expression in CD4$^+$ and CD8$^+$ effector cells. Data indicate that the relative expression of *Ifitm3* was higher in Teff cells and Treg (Supplementary Fig. 9a). Therefore, we hypothesized that IFITM3 also play a vital role in CD4$^+$ and CD8$^+$ T cells. To verify this, we crossed the *Ifitm3*$^{fl/fl}$ mice with *Cd4*$^{cre}$ mice to develop mice with T-cell-specific deletion of IFITM3. Phenotypic analysis shows that IFITM3 deficiency in CD4$^+$ T cells and CD8$^+$ T cells leads to elevated secretion of IFNγ after TCR stimulation (Supplementary Fig. 9b, c). The proliferation and apoptosis results indicated that the IFITM3-deficient T cells possess an increased proliferation rate (Supplementary Fig. 9d). In addition, we found the naïve CD4$^+$ T cells from KO mice preferred to differentiate into Th1 cells but not Treg cells, which were consistent with our findings in Treg conditional knock-out mice (Supplementary Fig. 9e, f).

We further inoculated WT and KO mice with MC38 murine colon cancer cells subcutaneously to determine the function of IFITM3 in vivo. The KO mice also had profound reductions in MC38 tumor size with enhanced secretion of effector cytokines (Supplementary Fig. 9g–j). These results indicated that besides the important role in TI-Treg, IFITM3 also participates in regulating CD4$^+$ and CD8$^+$ T cells. Therefore, IFITM3 and the regulation of IFITM3-related protein complex can be targeted to minimize the Treg function in the tumor microenvironment.

## Discussion

As a prominent subpopulation of tumor-infiltrating CD4$^+$ T cells, Tregs suppress anti-tumor responses and indicate a potential clinical target for tumor immunotherapy. Taking advantage of the fragility and plasticity of Treg cells, we can weaken their suppressive function and improve their inflammatory potential, which is helpful for anti-tumor immune response. This study uncovered a negative feedback loop that maintains the TI-Treg function to block anti-tumor response. We showed that TI-Treg cells can be targeted by perturbating the STAT1-IFITM3 loop.

Specifically, we found that IFITM3-deficient TI-Treg exhibits a demolished suppression function. Thus, *Ifitm3*$^{fl/fl}$*Foxp3*$^{YFP-Cre}$ mice exhibit enhanced anti-tumor immunity, and there is a higher population of tumor-killing cells with improved function. IFITM3 deficiency in Treg cells resulted in Th1-like unstable IFITM3-deficient Treg cells, accompanied by increased Th1 responses and vigorous activation. IFITM3 deficiency increased the IFNγ secretion of Treg cells and induced higher expression of ISGs, including STAT1. Meanwhile, as an amplifier of PI3K signaling at the cell membrane[16], IFITM3 could also suppress AKT phosphorylation, thereby maintaining FOXO1 localization in the nucleus to stabilize FOXP3 expression.

The role of IFN-γ in the tumor microenvironment is paradoxical, as it exerts both anti-tumor and tumor-promoting effects under certain conditions. A previous study showed that immune inhibitory anti-CTLA4 (Ipilimumab) treatment could increase IFNγ production by T cells—and IFNγ-related gene deficiency in tumor cells induced resistance to anti-CTLA4 immune checkpoint therapy[29]. In addition to tumor cells, IFNγ signaling culminates in immune cell activation. IFNγ exerts its downstream effects by binding to the IFNγ receptor (IFNGR) consisting of two subunits, IFNGR1 and IFNGR2. The binding of IFNγ to its receptors results in activating the Janus kinases, JAK1 and JAK2, and subsequent phosphorylation, dimerization, and activation of STAT1[30]. However, the specific regulatory effects of IFNγ and STAT1 signaling pathways in the anti-tumor immune response are still unknown. Our results showed that IFITM3 could competitively combine and bind to JAK1 and JAK2 on the cell membrane and suppress the phosphorylation of STAT1, thus suppressing the IFNγ secretion from Tregs and maintaining Tregs stability. As interferon-induced genes, STAT1 and IFITM3 were upregulated by IFNγ and other cytokines that induce Th1 response in the tumor microenvironment.

Treg cells mainly induced by TGF-β in the periphery are significantly inhibited by IFNγ in a STAT1-dependent manner[18], indicating that STAT1 may also play an important role in Treg differentiation and function. In STAT1-deficient Treg cells, we found the IFITM3 expression was reduced and could not be induced by IFNγ. These results proved that STAT1 and IFITM3 formed a feedback loop in TI-Treg cells to maintain the Treg function and stability. Upon blocking the IFNγ stimulation, the impact of IFITM3 on STAT1 was rescued, suggesting that the correlation of IFITM3 and STAT1 is dependent on IFNγ signaling.

In the mice tumor model, we observed that the deficiency of STAT1 or IFITM3 alone could induce fragile TI-Treg cells and thus strengthen anti-tumor immunity. However, depletion of both abrogated the enhancement of anti-tumor ability caused by the alterations of Treg fragility. Thus, we proposed that the STAT1-IFITM3 feedback loop could be a potential target when designing Treg-specific anti-tumor therapies. Since STAT1 could regulate many other downstream genes and IFITM3 may also influence the expression of many proteins after IFNγ stimulation, it is practical to disrupt this loop by eliminating these two genes in Treg cells.

In summary, we identified that the feedback loop of STAT1-IFITM3 maintains Treg function and stability in the tumor microenvironment, which provides innovative insights and targets to achieve anti-tumor immunity in clinical patients. Our data defines IFITM3-mediated Treg suppression function in a mice tumor model and highlights the correlation between STAT1 and IFITM3 regulation in TI-Treg cells. Since the STAT1-IFITM3 feedback loop depends on IFNγ, our research also provides innovative sights of targeting Treg perturbation to regulate anti-tumor immunity in the IFNγ-enriched tumor microenvironment (Fig. 8). Strategies that disrupt the balance of the STAT1-IFITM3 negative feedback loop but do not eradicate this loop may be broadly applicable in clinical anti-tumor therapy by inducing Treg dysfunction.

## Methods
### Human samples
Patient-derived PBMC, ESCA tissues, and matched normal tissues were obtained from Ruijin Hospital, and COAD tissues and matched normal tissues were obtained from Renji Hospital. The study protocol was approved by the ethics committee of Ruijin Hospital (2021-224) and Renji Hospital (KY2022-174-B) complied with all relevant ethics

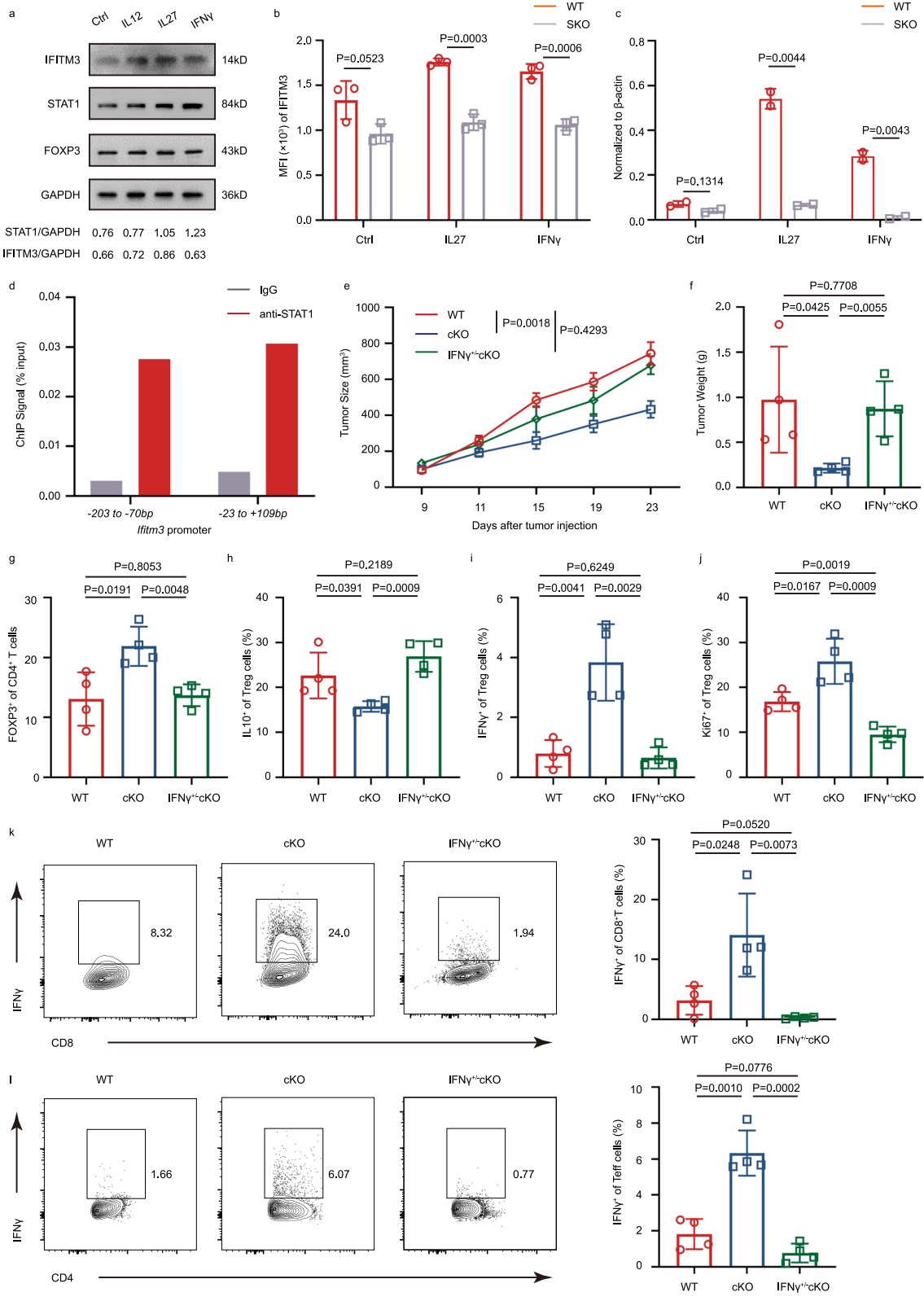

regulations, and informed consent was obtained from each patient. Solid and normal tissues were freshly isolated and digested for flow cytometry analysis.

## Mice

All mouse lines were on the C57BL/6J genetic background. *Ifng* KO mice were a gift from Gonghua Huang(Guangdong Medical University,

China). *Ifitm3^{fl/fl}* mice and *Stat1^{fl/fl}* mice were generated by Cyagen Bioscience using a LoxP-targeting system. *Foxp3^{YFP-cre}* mice were purchased from the Jackson Laboratory (stock number, 016959), which carries an internal ribosome entry site (IRES) and a yellow fluorescent protein (YFP) fused to a codon-optimized Cre recombinase sequence downstream of the internal stop codon of the Foxp3 gene. And *Cd4^{cre}* mice were also purchased from the Jackson Laboratory (stock number,

**Fig. 6 | IFITM3 and STAT1 form a feedback loop and it is IFNγ dependent.**
**a** Immunoblot analysis of IFITM3, STAT1, and FOXP3 in Treg cells after IL12 (50 ng/ml), IL27 (50 ng/ml), and IFNγ (50 ng/ml) treatment for 24 h. **b** Flow cytometric analysis of IFITM3 in WT and SKO Treg cells after IL12, IL27, and IFNγ treatment for 24 h ($n = 3$). **c** RT–qPCR analysis of IFITM3 mRNA level in WT and SKO Treg cells after IL12, IL27, and IFNγ treatment for 24 h ($n = 3$). **d** ChIP-qPCR analysis of −203 to −70bp and −23 to +109 bp of exon1 of *Ifitm3* that is regulated by STAT1 in WT Treg cells. Data in d are representative of 3 independent experiments. **e** Tumor growth in WT, cKO, and IFNγ$^{+/-}$ cKO mice injected s.c. with MC38 murine colon cancer cells (WT: $n = 7$; cKO: $n = 7$; IFNγ $^{+/-}$ cKO: $n = 7$). **f** Tumor weight of WT, cKO, and IFNγ$^{+/-}$ cKO mice injected s.c. with MC38 murine colon cancer cells ($n = 4$ per group).

**g** Percentage of Treg cells in tumor of WT, cKO, and IFNγ $^{+/-}$cKO mice injected s.c. with MC38 murine colon cancer cells (day 23, $n = 4$ per group). **h, i** Frequency of IL10 (**h**) and IFNγ (**i**) producing Treg cells in tumor of WT and cKO mice injected s.c. with MC38 murine colon cancer cells (day 23, $n = 4$ per group). **j** The quantification of the MFI of Ki67 in Treg cells in tumors of WT, cKO, and IFNγ $^{+/-}$cKO mice injected s.c. with MC38 murine colon cancer cells (day 23, $n = 4$ per group). **k, l** Cytokine secretion of CD8$^+$ (**k**) and CD4$^+$ T cells (**l**) in tumor of WT, cKO, and IFNγ$^{+/-}$ cKO mice injected s.c. with MC38 murine colon cancer cells (day 23, $n = 4$ per group). The data **f–l** are presented as representative plots and as summary graphs ($n = 4$ per group). Data are represented as the mean ± SD. Source data are provided as a Source Data file.

022071). All mice were allowed free access to food and water and were housed under 12 h light and dark cycles at room temperature (22–26 °C) with relative humidity around 40%. All mice were maintained in a specific–pathogen–free (SPF) facility at the Shanghai Jiao Tong University School of Medicine. Male and female mice were sex-matched and used at 6 to 8 weeks of age. All animal experiments followed the protocols approved by the Institutional Animal Care and Use Committee at the Institute of Shanghai Immunology, School of Medicine, Shanghai Jiao Tong University under protocol number A-2022-031.

## Cell culture
HEK293T cells (ATCC) and MC38 murine colon cancer cells (ATCC) were cultured in DMEM medium supplemented with 10% fetal bovine serum (FBS) and 100 units/ml penicillin-streptomycin. Isolated mouse and human T cells were cultured in RPMI 1640 medium supplemented with 10% FBS, 100 units/ml penicillin-streptomycin, 1% Sodium pyruvate, and 1% non-essential amino acids.

## T-cell isolation and stimulation
To enrich lymphocytes from different tissues like the lung, liver, colon, subcutaneous fatty tissue (SAT), and visceral adipose tissue (VAT), we first cut the tissue into small pieces. Lung, liver, SAT, and VAT were digested for 30–60 min at 37 °C with Collagenase D (1 mg/ml, Roche), DNaseI (150 μg/ml, Roche), and 10% FBS in RPMI 1640 medium. Colon was digested for 60 min at 37 °C with Collagenase VIII (1 mg/ml, Roche), DNaseI (150 μg/ml, Roche), and 10% FBS in RPMI 1640 medium. MC38 tumors were digested for 30 min at 37 °C with Collagenase IV (1 mg/ml, Roche), DNaseI (150 μg/ml, Roche), and 10% FBS in RPMI 1640 medium. Then, digested tissues were strained through 40 μm cell strainers and subjected to flow cytometric analysis.

## In vitro suppression assay
Lymphocytes were isolated from the spleen and peripheral lymph nodes, and then CD4$^+$ T cells were enriched by using CD4 (L3T4) Microbeads (130-049-201, Miltenyi Biotec). Then CD4$^+$CD25$^{hi}$YFP$^+$ Treg cells and CD4$^+$CD25$^-$YFP$^-$ Tconv cells were sorted on a BD FACS Aria II cell sorter (BD Biosciences). Treg cells and CTV (C34571, Thermo Fisher)-labeled Tconv cells were cocultured at the ratio of 1:1, 1:2, 1:4, and 1:8 with the stimulation of mouse anti-CD3/CD28 beads (11453D, Thermo Fisher). And then analyzed proliferation as determined by CTV dilution and the cytokine secretion by flow cytometry.

## CD4$^+$ T-cell subsets in vitro polarization
CD4$^+$CD25$^-$YFP$^-$CD62L$^{hi}$ naïve CD4$^+$ T cells were sorted on a BD FACS Aria II cell sorter. Naive CD4$^+$ T cells were stimulated with anti-CD3/CD28, rhIL-2 (200-02-1 mg, Peprotech, 50 U/ml), rmIL-12 (419-ML-010/CF, R&D, 10 ng/ml), and anti-IL-4 (404135, Biolegend, 10 μg/ml) to differentiate into Th1 cells. CD4$^+$ T cells were stimulated with anti-CD3 (BE0001-1, BioXCell, 2 μg/ml), anti-CD28 (BE0015-1, BioXCell, 2 μg/ml), rhIL-2 (50 U/ml), and rmTGF-β (7666-MB-005/CF, R&D, 5 ng/ml) to differentiate into iTreg cells. The differentiation efficiency of iTreg cells was evaluated by the percentage of FOXP3 in CD4$^+$ T cells, and the

differentiation efficiency of Th1 cells was evaluated by the percentage of IFNγ in CD4$^+$ T cells. The cytokines used in Treg in vitro experiments were Recombinant Mouse IL27 (577402, Biolegend, 50 ng/ml), Recombinant Mouse IL12 Protein (419-ML-010, R&D, 50 ng/ml), Recombinant Mouse IFN-gamma (485-MI-100/CF, R&D, 50 ng/ml)

## Coculture of MC38 and splenocytes in vitro
MC38 cells ($5 \times 10^4$ cells/ml) were added to a 48-well plate 12 h before coculture. Splenocytes ($5 \times 10^5$ cells/ml) were inoculated into each well with MC38 cells to achieve a ratio of 1:10 of MC38 cells to splenocytes ($n = 4$). Following 48 h of coculture, the cell supernatant was collected for IFNγ ELISA analysis (430804, Biolegend).

## Flow cytometry and cytokines production analysis
All flow cytometry data were collected from LSRII (BD) flow cytometry and were analyzed using FlowJo software. For surface marker staining, cells were incubated with specific antibodies in PBS containing 2% FBS plus 2 mM EDTA, including Viability Dye (65-0865-14, eBioscience, 1:1000), anti-CD4 (45-0042-82, eBioscience, 1:300), anti-CD8 (563898, BD Pharmingen, 1:300), anti-CD25 (102008, Biolegend, 1:300), anti-CD45.2 (109832, Biolegend, 1:300), anti-CD44 (130-102-606, Miltenyi Biotec, 1:200), anti-CD62L (20-0621-U100, Tonbo, 1:200), anti-NK1.1 (11-5941-85, eBioscience, 1:200) for 30 min on ice. Cells were then washed with ice-cold PBS twice and analyzed by flow cytometry. For intracellular marker staining, cells were fixed, permeabilized, and then labeled with antibodies, including anti-FOXP3 (11-5773-82, eBioscience, 1:300), anti-IL-17A (130-102-344, Miltenyi Biotec, 1:200), anti-IFNγ (17-7311-82, eBioscience, 1:200), anti-Ki67 (25-5698-82, eBioscience, 1:200), anti-IL10 (12-7101-82, eBioscience, 1:200), anti-TNFα (12-7321-82, eBioscience, 1:200), and anti-pSTAT1 (666404, Biolegend, 2ul/test). For cytokine expression detection, cells were stimulated with phorbol 12-myristate 13-acetate (P1585, Sigma), ionomycin (I3909, Sigma), and Golgi Stop (554724, BD Bioscience) for 4–6 h. After stimulation, cells were fixed and permeabilized, followed by staining with cytokine antibodies. The gating strategy is provided in Supplementary Fig. 10.

## Plasmids and cytokines
Myc-tagged IFITM3 and all IFITM3 mutations were subcloned into the PIP-Myc vector, and HA-tagged STAT1 was subcloned into the PIP-HA vector. Flag-tagged JAK1 and HA-tagged JAK2 were subcloned into PIP-Flag and PIP-HA vectors, respectively.

## Immunoblotting and immunoprecipitation
Cells were washed with ice-cold PBS and lysed on ice for 30 min in RIPA lysis buffer (150 mM NaCl, 5% Glycerin, 1% Triton X-100, 50 mM Tris-HCl, pH7.5). For each immunoprecipitation reaction, 0.2 ml of the cell lysate was incubated with 1 μg of antibody (anti-Myc, 9E10, Santa Cruz, sc-40; anti-HA, H6908, Sigma-Aldrich) at 4 °C for 4 h, and then incubated with 10 μl of protein A/G agarose beads at 4 °C for 1 h. The sediments were then subjected to SDS-PAGE and immunoblot analysis. The antibodies used in immunoblotting were as follows: anti-Myc (9E10, Santa Cruz,1:2000), anti-HA (H6908, Sigma-Aldrich, 1:2000), anti-Flag (F3165, Sigma-Aldrich, 1:15000), anti-LC3B (ab192890,

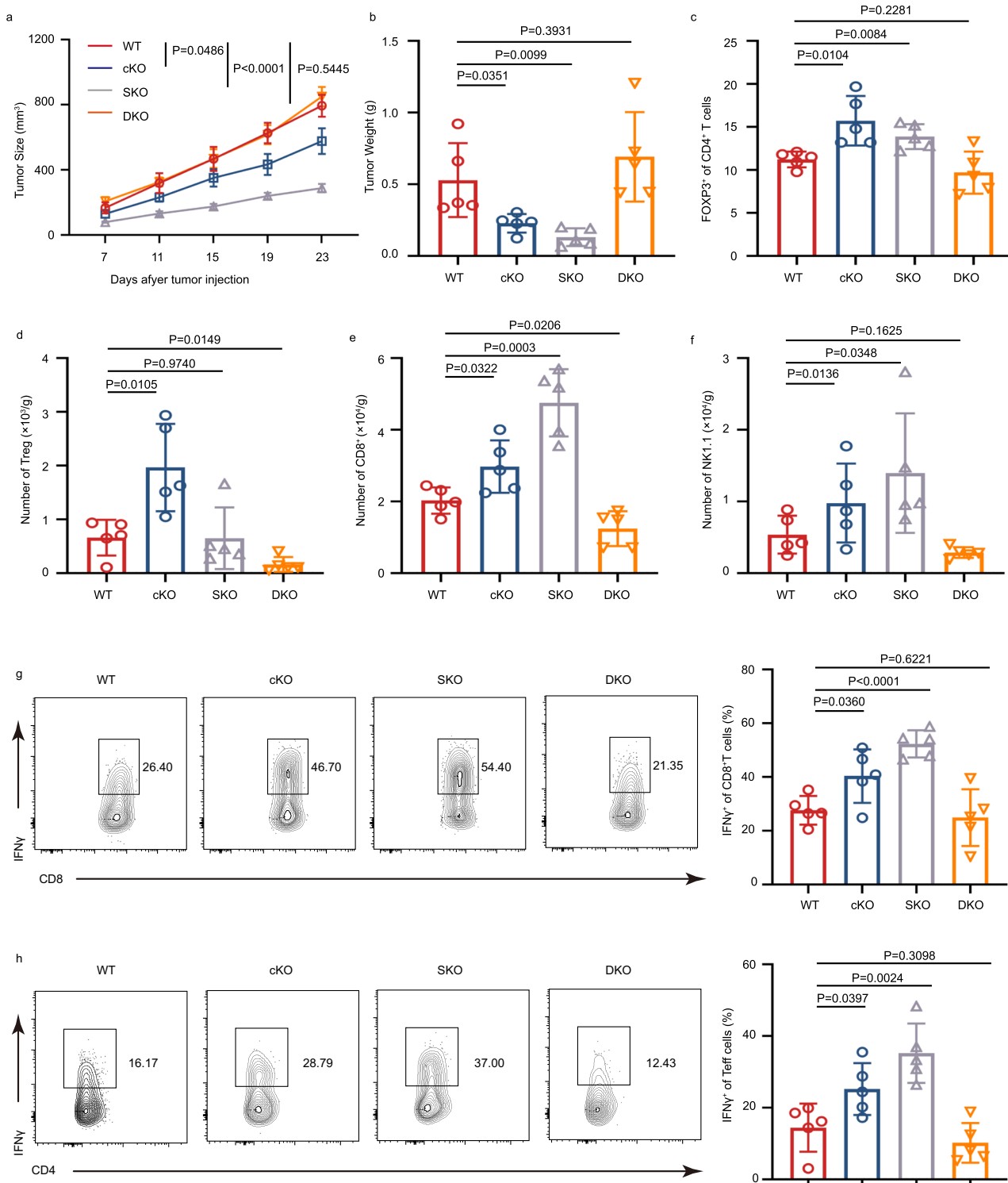

**Fig. 7 | STAT1-IFITM3 feedback loop plays an essential role in tumor-infiltrating Tregs. a** Tumor growth in WT, cKO, SKO, and DKO mice injected s.c. with MC38 murine colon cancer cells (WT: $n = 7$; cKO: $n = 7$; SKO: $n = 7$; DKO: $n = 7$). **b** Tumor weight of WT, cKO, SKO, and DKO mice injected s.c. with MC38 murine colon cancer cells ($n = 5$ per group). **c** Percentage of Treg cells in tumor of WT, cKO, SKO, and DKO mice injected s.c. with MC38 murine colon cancer cells ($n = 5$ per group). **d**–**f** Cell number of Treg cells (**d**), CD8[+] cells (**e**), and NK cells (**f**) in tumor of WT, cKO, SKO, and DKO mice injected s.c. with MC38 murine colon cancer cells (Day 23, $n = 5$ per group). **g**, **h** Cytokine secretion of CD8[+] (**g**) and CD4[+] T cells (**h**) in tumor of WT, cKO, SKO, and DKO mice injected s.c. with MC38 murine colon cancer cells (Day 23, $n = 5$ per group). The data **b**–**h** are presented as representative plots and as summary graphs ($n = 5$ per group). Data are represented as the mean ± SD. Source data are provided as a Source Data file.

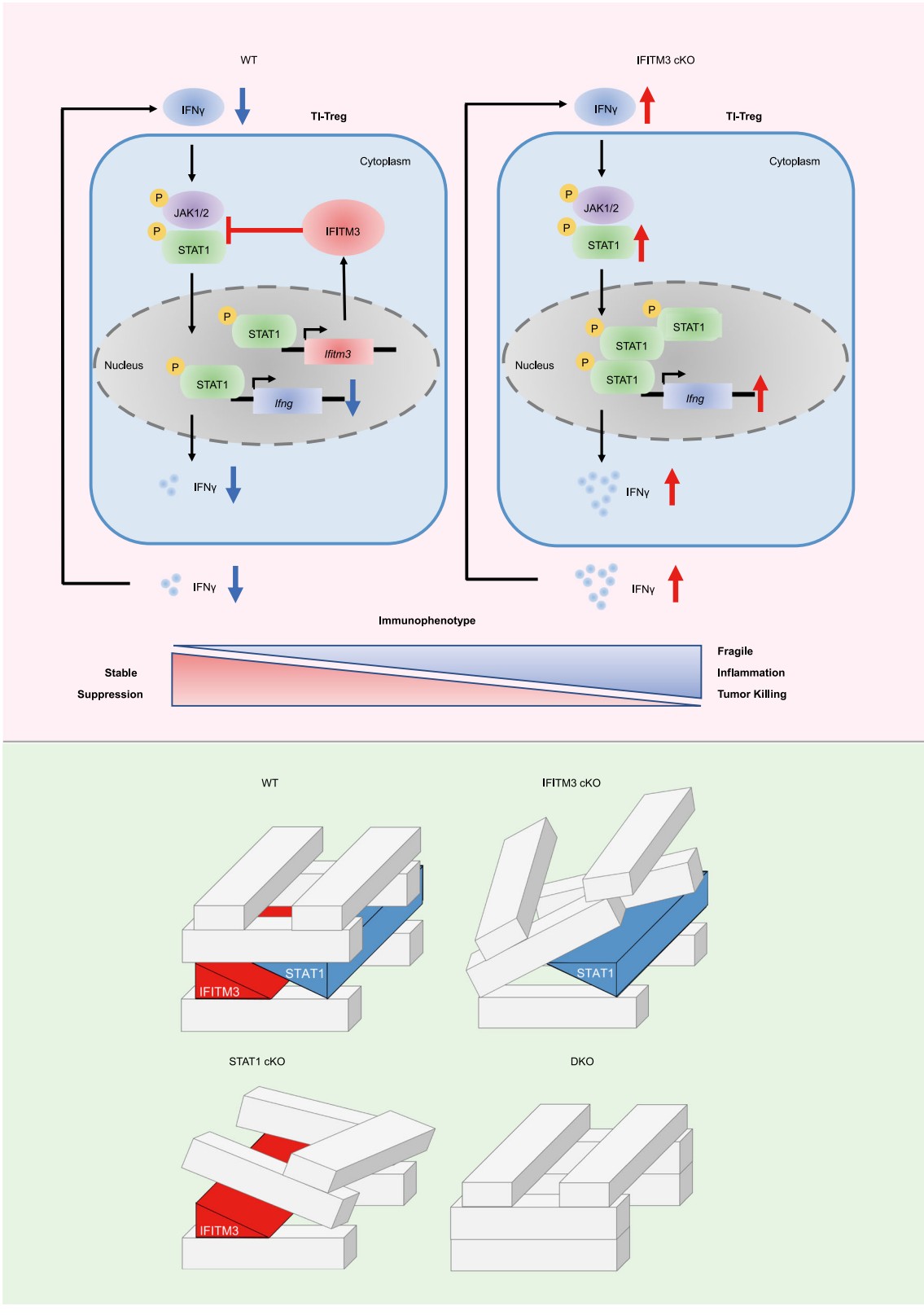

Abcam,1:1000), anti-IFITM3 (11714-1-AP, Proteintech,1:5000), anti-STAT1 (14994T, Cell Signaling Technology, 1:1000), anti-pSTAT1 (Tyr701) (7649T, Cell Signaling Technology, 1:1000), anti-FOXP3 (14-7979-82, eBioscience, 1:3000), anti-FOXO1 (2880T, Cell Signaling Technology, 1:1000), anti-Phospho-FoxO1 (Thr24)/FoxO3a (Thr32) (9464T, Cell Signaling Technology, 1:1000), anti-AKT (60203-2-Ig, Proteintech, 1:5000), anti-p-AKT (9271T, Cell Signaling Technology, 1:1000), anti-GAPDH (60004-1-Ig, Proteintech, 1:8000), anti-β-Actin (66009-1-Ig, Proteintech, 1:8000), anti-LaminB (66095-1-Ig, Proteintech, 1:8000).

**Fig. 8 | Model explaining our findings.** A STAT1-IFITM3 negative feedback loop limits the IFNγ-signaling pathway to maintain Treg function and stability. In WT Treg cells, IFNγ induces the phosphorylation of STAT1, and then the phosphorylated STAT1 transfer to the nucleus and induce the transcription and translation of IFITM3. And then IFITM3 will suppress the phosphorylation of STAT1 and lead to the degradation of STAT1 to inhibit the further activation of the IFNγ-signaling pathway (WT Treg on the left). In IFITM3-deficient Treg cells, the phosphorylation of STAT1 and IFNγ-signaling pathway are strongly activated, thus Treg cells show upregulated IFNγ signaling and revert to inflammatory cells (IFITM3 cKO Treg on the right). We simulate TI-Treg cells as buildings made up of blocks and simulate IFITM3 and STAT1 as two blocks balancing each other to maintain the stability of the buildings. When we take away block-IFITM3 (IFITM3 cKO) or block-STAT1 (STAT1 cKO), the building loses balance and collapses. But when we take away both block-IFITM3 and block-STAT1 (IFITM3 and STAT1 DKO), the building remains in normal equilibrium. So this block model greatly simulates Treg function and stability in WT, IFITM3 cKO, STAT1 cKO, and DKO mice.

## Cell nuclear/cytoplasmic separation

Cells were washed with ice-cold PBS and lysed in buffer B (10 mM KCl, 0.1 mM EDTA, 0.5 mM NP-40, 10 mM Hepes) on ice for 10 min. Then the cell lysate was centrifuged at 4600 g at 4 °C for 30 s. The supernatant was the cytoplasmic protein, and the sediment was then lysed in buffer C (400 mM NaCl, 1 mM EDTA, 20 mM HEPES) and vortexed at 4 °C for 1 h. Then, the cell lysate was centrifuged at $13,000 \times g$ at 4 °C for 2 min, and the supernatant was the cell nuclear protein.

## Tumor models and treatment

MC38 murine colon cancer cells were injected s.c. into the flank region of 6- to 8-week-old mice ($5 \times 10^5$ cells per mouse). To minimize individual variations, 8 to 10 age- and sex-matched mice in each group were used. The tumor-challenged mice were monitored for tumor size (tumor area indicates tumor size). Mice were killed after 23 days, and tumors were collected, weighed, and examined histologically. Tumor size did not exceed 20 mm in any direction, and tumor volume did not exceed 2000 mm³. Tumor-induced lethality was defined as a tumor size reaching 225 mm². Anti-mouse-IFNγ (100ug/injection, BE0055, BioXcell) and isotype control (100 μg/injection, BE0088, BioXcell,) antibodies were intraperitoneally injected at day4, day8, day12, and day16.

## RT−qPCR and RNA-seq analysis

Total RNA was extracted using TRIzol reagent (15596018, Ambion). cDNA was synthesized by a reverse transcriptase kit (R323-01, Vazyme), followed by qPCR analysis (SYBR Green; Q711-02, Vazyme) on an ViiA7 PCR system (Applied Biosystems). β-Actin was used as an internal control for normalization. Primers used for RT−qPCR were listed in Supplementary Table 1.

Tumor-infiltrating WT and IFITM3-deficient Treg cells were isolated, and all the samples were submitted to BGI Genomics (BGI Group, Shenzhen, China) for RNA extraction and mRNA sequencing (BGI DNBSEQ RNA-Seq platform). Significantly differentially expressed genes were defined by fold change ≥2 and *P*-value < 0.05. Heatmap plots were generated with the R package 'pheatmap'.

## ChIP assay and RT−qPCR

For ChIP sample preparation, $1 \times 10^7$ iTreg cells per sample were cross-linked, and the chromatin was digested as described in the SimpleChIP® Plus Enzymatic Chromatin IP Kit (9005, Cell Signaling Technology) manual. 8–10 μg chromatin in each sample was immunoprecipitated with STAT1 Monoclonal Antibody (AHO0832, Invitrogen, 5ul/test) or Mouse IgG1 kappa Isotype Control (14-4714-82, eBioscience, 5ul/test) overnight, followed by precipitation with ChIP-Grade Protein G Magnetic Beads (9006S, Cell Signaling Technology). Primers used for ChIP-RT−qPCR were listed in Supplementary Table 2.

## Immunofluorescent staining

HEK293T cells and Hela cells were grown on glass coverslips were fixed in 4% paraformaldehyde for 10 min at room temperature and then permeabilized with 0.1% Triton X-100 (in phosphate-buffered saline) for 10 min at 4 °C. Then, cells were blocked with 1% BSA (in phosphate-buffered saline) for 30 min and then incubated with the primary antibodies for 1 h at room temperature. Cells were then washed with PBS and stained with Alexa Fluor-labeled anti-mouse (A32727, Invitrogen, 1:1000), anti-rabbit secondary antibodies (A227034, Invitrogen, 1:1000) for 30 min at 4 °C in the dark, and washed with PBS. Finally, labeled cells were counterstained with DAPI for 10 min, and images were captured using a Leica SP8 confocal laser scanning microscope.

## Statistics

*P*-values were calculated with an unpaired t-test, GraphPad Prism 8.0, as specified in figure legends. All data represent means ± SD unless stated otherwise.

## Reporting summary

Further information on research design is available in the Nature Portfolio Reporting Summary linked to this article.

## Data availability

The gene expression of COAD and ESCA patients are downloaded from The Cancer Genome Atlas (TCGA) (https://cancergenome.nih.gov/). The RNA-sequencing data of tumor-infiltrating Treg cells are available on the Gene Expression Omnibus public database (GEO) database under accession code GSE248223. Source data are provided in this paper. Information required for reanalyzing data from this paper is available from the corresponding author upon request. Source data are provided in this paper.

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

## Acknowledgements
Our research is supported by the National Key Research and Development Project (2019YFA0906102, 2021YFC2500900); National Natural Science Foundation of China (81830051, 32130041, 82241222, 32170925, 82372855, 82072557); Shenzhen Science and Technology Program (KQTD20210811090115019); Innovative research team of high-level local universities in Shanghai (SHSMU-ZDCX20210601); Shanghai Frontiers Science Center of Cellular Homeostasis and Human Diseases. We thank members of Bin Li's laboratory for their kind help and suggestions. We also thank for the support from the Core Facility of Basic Medical Sciences, Shanghai Jiao Tong University School of Medicine, the Core Facility of Basic Medical Sciences, and the sequencing core at Shanghai Institute of Immunology, Shanghai Jiao Tong University School of Medicine.

## Author contributions
F.P., H.C.L. and B.L. designed the study; X.L. and H.C. performed the experiments; X.L., W.Q.Z., H.C., Q.L. analyzed and interpreted the data; X.L., W.Q.Z. and Y.H. wrote and revised the manuscript. S.K., Y.H., W.Y.Z. and H.C.L. collected clinical samples. F.Z., Y.L., L.X. and X.D. reviewed the manuscript and gave conceptual advice; C.W., G.H., B.S., Q.Z., H.B.L., L.L. and X.T. provided technical and material support. All authors approved the manuscript.

## Competing interests
B.L. is a cofounder of Biotheus Inc. and chairman of its scientific advisory board. The other authors declare no competing interests.

## Additional information

[1]Center for Immune-Related Diseases at Shanghai Institute of Immunology, Department of Immunology and Microbiology, Shanghai Jiao Tong University School of Medicine, Shanghai, China. [2]Department of Thoracic Surgery, Ruijin Hospital, Shanghai Jiao Tong University School of Medicine, Shanghai, China. [3]Center for Cancer Immunology Research, Shenzhen Institute of Advanced Technology, Chinese Academy of Sciences, Shenzhen, Guangdong, China. [4]Department of Biochemistry and Molecular Cell Biology, Shanghai Key Laboratory for Tumor Microenvironment and Inflammation, Key Laboratory of Cell Differentiation and Apoptosis of Chinese Ministry of Education, Shanghai Jiao Tong University School of Medicine, Shanghai, China. [5]Department of Gastrointestinal Surgery, Ren Ji Hospital, Shanghai Jiao Tong University School of Medicine, Shanghai, China. [6]Department of Microbiology, School of Medicine, University of Alabama at Birmingham, Birmingham, AL, USA. [7]Key Laboratory of Metabolism and Molecular Medicine of the Ministry of Education, Department of Biochemistry and Molecular Biology of School of Basic Medical Sciences, Shanghai Medical College of Fudan University, Shanghai, China. [8]Department of Medical Oncology, Harbin Medical University Cancer Hospital, Harbin, Heilongjiang, China. [9]Institute of Arthritis Research, Guanghua Integrative Medicine Hospital, Shanghai University of Traditional Chinese Medicine, Shanghai, China. [10]Department of Obstetrics and Gynecology, Shanghai Jiao Tong University School of Medicine Affiliated Renji Hospital, Shanghai, China. [11]Guangdong Provincial Key Laboratory of Medical Molecular Diagnostics, Guangdong Medical University, Dongguan, Guangdong, China. [12]Shanghai Immune Therapy Institute, Shanghai Jiao Tong University School of Medicine Affiliated Renji Hospital, Shanghai, China. [13]Department of Thoracic Surgery, Shanghai Pulmonary Hospital, Tongji University, Shanghai, China. [14]Department of Oncology, Department of Hepatobiliary Surgery, The First Affiliated Hospital of Anhui Medical University, Hefei, China. [15]Department of Integrated TCM & Western Medicine, Shanghai Skin Disease Hospital, School of Medicine, Tongji University, Shanghai, China. [16]These authors contributed equally: Xinnan Liu, Weiqi Zhang, Yichao Han, Hao Cheng, Qi Liu. [17]These authors jointly supervised this work: Fan Pan, Hecheng Li, Bin Li. ✉e-mail: fan.pan@siat.ac.cn; lihecheng2000@hotmail.com; binli@shsmu.edu.cn

