## [Peer Review File · Nature Communications]

FOXP3+ regulatory T Cell perturbation mediated by the IFN γ -STAT1-IFITM3 feedback loop is essential for anti-tumor immunityREVIEWER COMMENTS

Reviewer #1 (Remarks to the Author):

Regulatory T cells accumulate in the tumour microenvironment and inhibit the antitumor response, allowing tumour growth. Therefore, any knowledge about the function, development or stability of these cells in the TME is an opportunity to improve current therapies and therefore of great interest.

The authors focus on demonstrating the role of IFITM3-STAT1 in the stability of regulatory T cells. They observe that inhibition of this axis destabilises FOXP3 and its suppressor activity, increasing the expression of ISGs and leading to an increased anti-tumour response with an impact on tumour growth. The results are novel, as they describe a new molecular mechanism of Treg phenotype differentiation and maintenance in the tumour microenvironment.

My main concern is about the cell-intrinsic role of both KO models, IFITM3 and STAT1. The authors should demonstrate that at the time of the assays the knock-out of these two molecules are not affecting the Treg resting state, they did a study in naive mice for the IFITM3 KO and in 12-month old mice there is an increase of effector T cell and IFN γ positive cell. Besides, any data are shown about STAT1 $^{fl/fl}$ Foxp3 Cre animals. On the other hand, the part of the role of IFN γ in the feedback loop of IFITM3-STAT1 should be reinforced with further studies.

Another weakness of the article is the lack of specific anti-tumour response studies, either towards the MC38 tumour line or against the peptide MuLv $_{p15E}$ KSPWF $_{TTL}$ immunodominant epitope of the murine retrovirus envelope. The author should study the frequency of specific tetramers, and the specific anti-tumour response to the MuLv $_{p15E}$ KSPWF $_{TTL}$ epitope or MC38 cell line.

In relation to the methods, I would highlight two points to be specified. The strains of mice used need to be better described, they make several crosses and should be better detailed. On the other hand, it is not clear what type of RNASeq was used. Also, how old are the animals treated? The author should show detail this.

The paper is well written and organized. However, there are some points that should be addressed before its publication.

Major comments

1. My main concern is in relation to the genetic models the authors are working with in their study, as they are depleting their targets, IFITM3 or STAT1 in regulatory T cells from ontogeny, if these targets have an impact on the development or function of regulatory T cells it may generate results due to extrinsic cells that would mask the real results. The authors should

First analyse the phenotype (both percentage and number) of T populations in thymus, bone marrow, lymph nodes and spleen at the age of the studies and in the long term.

In suppl figure 2, how old are the animals under study? The author just shown the total percentage of CD4, CD8 or Treg but not their phenotype. The same study carry out on Spl Figure 3 has to be done in young mice in both KO models (IFITM3 and STAT1).

Another interesting option would be to work with inducible KO models or to generate mixed bone marrow chimeras, using a CD45.1/2 or CD90.1/2 reporter. Reconstitute C57BL/6 mice with 1:1 mixtures of congenically distinct CD45.1+ wild-type (WT) and CD45.2+ IFITM3 or STAT1 $^{-/-}$ mature lineage-depleted (Lin $^{-}$) bone marrow (BM) cells. (Roychoudhuri R et al, Nat Immunol, 2016. Doi : doi: 10.1038/ni.3441)

be sure of the cell-intrinsic function of IFITM3/STAT1 in Treg.

2. To demonstrate the impact of IFN γ in the feedback loop of IFITM3 and STAT1, the

authors work with a IFN γ +/- IFITM3^{fl/fl} animal and the results are shown in Figure 6e-l. As far as I understand the authors work with heterozygous for the non-conditional KO IFN γ mice, to confirm these results it would be necessary to repeat the same analysis but with a different technical approach.; such as anti-hIFN- γ neutralizing mAb

3. The authors should measure the antitumor specific response in their in vivo studies, regarding to KO IFITM3, KO STAT1, or double KO,

Ex vivo stimulation of splenocytes against MC38 cell line, ELISPOT, ELISA should do to measure IFN γ or other cytokines

Parameters such as: number/mg tumor of tetramer positive (peptide MuLv,p15E KSPWF^TTLL), phenotype of T cell (CD44, CD262L, PD1, TM3, CD137, ICOS) both tumor specific and unspecific should be measure within the tumour and periphery

Minor comments

1. Figure 1, in MC38 murine model, comparison of peripheral Treg vs intratumor Treg, to demonstrated the role of IFITM3 is depend on tumour microenvironment

2. Suppl 2d. The author should show the numbers of DP, DN, CD4 and CD8. Not only the percentage

3. Figure 2, Boolean analysis has to be done to measure the T polyfunctionality instead a single positive analysis

4. Suppl 3. At what month the naïve animals start to increase de effector T cell and IFN γ + cell?

5. In conditional KO models, is there any compensatory mechanism? How is the expression of IFITM10, IFITM2, IFITM1 on KO Treg cells?

6. Figure 2, shown Survival curve

7. Test the suppression function of KO Treg in ex vivo experiments. Isolate de Treg from KO conditional mice (IFITM3 and STAT1) and coculture with effector T cells stimulate with CD3/CD28, analyses of cytokine production or proliferation rate of effector cell.

8. Suppl Fig4, validation for RNASeq results should be done in intratumor Treg

9. Some western blots throughout the article are not quantified.

10. Suppl Fig 5a, b. Demonstrate the interaction STAT1, IFITM3 with LC3B on Treg, not just in HEK 293 cell line

11. Figure 5d.e. Demonstrate the interaction STAT1, JAK1 /JAK2 on Treg, not just in HEK 293 cell line

Reviewer #2 (Remarks to the Author):

IFITM3 has been shown to play a complex role in tumorigenesis of cancer cells. However, the role and regulation of IFITM3 protein in tumor infiltrating regulatory T cells (TI-Tregs) is poorly understood.

The authors of this work have clearly characterized the effects of IFITM3-deficiency on Treg cells and described an IFN γ -dependent feedback loop governing the interactions of STAT1 with IFITM3. Crucially, IFITM3 conditional knockout Tregs were shown to lose their tumor-protective functions, such as IL10 or TGF β production, in favor of increased IFN γ expression, thus promoting anti-tumor responses.

Major points:

1. Although this paper focused on Tregs, demonstrating the positive regulation of IFITM3 by STAT1 has already been suggested in HCC cells (in <https://doi.org/10.1002/2211->

5463.12479). Furthermore, this same publication shows that IFITM3 can increase STAT1 levels in HCC cells, directly opposing what seems to be case in Tregs. The authors could address this and attempt to reconcile these contrasting observations.

2. The graph in figure 5h could use a legend that indicates statistical significance. In addition, it could be worth exploring why pAKT (at the 473 serine) levels display higher values initially in cKO mice should the difference at t=0 be significant, as this is inconsistent with the published results showing IFITM3 is a PIP3 scaffold to amplify PI3K signaling in B cells.
3. Whether STAT1 affects Treg stability (Foxp3, Ifng expression or suppressive assay) should be determined. In addition, Treg cell stability needs also to be checked in DKO mice.
4. The expression of IFITM3 in CD4+ effective T cells and CD8+ CTLs in the tumor should also be determined. And the potential functions of IFITM3 in these cells should be discussed.

Minor points:

1. In the last paragraph of the introduction, when referring to IFITM3 levels, “deregulated” could be substituted for another term that better denotes a deficiency as it can be interpreted as an overexpression.
2. Why were no co-immunoprecipitation experiments done to confirm the interaction of IFITM3 with JAK1 and JAK2 as it was the case for STAT1?

Reviewer #3 (Remarks to the Author):

This manuscript presents a very important study with a large amount of data that largely support the main conclusions. This work carries substantial impact on our understanding of the complexity of the immunosuppressive function and fragility of Treg cells in the tumor microenvironment. However, there are a few minor concerns that need to be addressed to improve the clarity and rigor of this work.

1. What are specifically the matched normal tissues used in comparison to tumor tissues in Fig 1? Are they isolated from the same tumor patients or healthy control subjects?
2. It appeared that the in vivo tumor growth results shown in Fig 6e-f and Fig 7a-b were performed only once with small number of mice per group (n=4 or 5). As these results are essential for the respective conclusions, replicate experiments should be performed to confirm the phenotypes.
3. There are many sentences in the manuscript that are problematic in grammar or not accurate scientifically. For example, the first sentence in the results section, “Tumor-infiltrating Treg cells feature correlation expression of IFITM3 and FOXP3”.....the first sentence in the last paragraph of discussion section, “In summary, we identified the feedback loop of STAT1-IFITM3 maintained Treg function and stability”. The first authors and correspondence authors should carefully go over the whole manuscript to make all necessary corrections.

REVIEWER COMMENTS

Reviewer #1 (Remarks to the Author):

Regulatory T cells accumulate in the tumour microenvironment and inhibit the antitumor response, allowing tumour growth. Therefore, any knowledge about the function, development or stability of these cells in the TME is an opportunity to improve current therapies and therefore of great interest.

The authors focus on demonstrating the role of IFITM3-STAT1 in the stability of regulatory T cells. They observe that inhibition of this axis destabilizes FOXP3 and its suppressor activity, increasing the expression of ISGs and leading to an increased anti-tumour response with an impact on tumour growth. The results are novel, as they describe a new molecular mechanism of Treg phenotype differentiation and maintenance in the tumour microenvironment.

My main concern is about the cell-intrinsic role of both KO models, IFITM3 and STAT1. The authors should demonstrate that at the time of the assays the knock-out of these two molecules are not affecting the Treg resting state, they did a study in naive mice for the IFITM3 KO and in 12-month old mice there is an increase of effector T cell and IFN γ positive cell. Besides, any data are shown about STA1Tfl/fl Foxp3Cre animals. On the other hand, the part of the role of IFN γ in the feedback loop of IFITM3-STAT1 should be reinforced with further studies.

Response: We have analyzed the immune homeostasis of STAT1 conditional knock-out mice and double knock-out mice, which indicated that the knock-out of these two molecules did not affect the Treg resting state. As for the role of IFN γ in the feedback loop of IFITM3-STAT1, we further used anti-IFN γ in the tumor model and made a conclusion consistent with the IFN γ knock-out mice. The supporting data has been introduced in more details in our point-to-point response.

Another weakness of the article is the lack of specific anti-tumour response studies, either towards the MC38 tumour line or against the peptide MuLv,p15E KSPWF TTL immunodominant epitope of the murine retrovirus envelope. The author should study the frequency of specific tetramers, and the specific anti-tumour response to the MuLv,p15E KSPWF TTL epitope, or MC38 cell line.

Response: We apologize for this weakness in our study. To address this weakness, we have performed a specific analysis of the anti-tumor response by coculturing tumor cells and splenocytes. The supporting data has been introduced in more details in our point-to-point response.

In relation to the methods, I would highlight two points to be specified. The strains of mice used need to be better described, they make several crosses and should be better detailed. On the other hand, it is not clear what type of RNASeq was used. Also, how old are the animals treated? The author show detail this.

Response: Thanks for highlighting this. We have described the mouse strains and our RNASeq data in more detail. We crossed the *Ifitm3*^{fl/fl} with *Foxp3*^{YFP-cre} mice, and *Stat1*^{fl/fl} mice with *Foxp3*^{YFP-cre} mice to get IKO mice and SKO mice, respectively. We crossed the *Ifitm3*^{fl/fl} with *Stat1*^{fl/fl} mice, and then with *Foxp3*^{YFP-cre} mice to get DKO mice. For *Ifng* KO mice, we crossed *Ifng*^{+/-} mice with WT and IKO mice. As for the RNASeq, we used low-input mRNA Library Preparation in the BGI DNBSEQ RNA-Seq platform (BGI-NGS-JK-RNA-007). The mice used in our manuscript were 6-8 weeks, except for the old mice that were 12 months old in Supplementary Figure 2.

The paper is well written and organized. However, there are some points that should be addressed before its publication.

Response: We thank you for recognizing the novelty and significance of our work that the feedback loop of IFITM3-STAT1 maintained Treg function and stability in the tumor microenvironment. We have responded and supplied the additional data following your comments. The point-to-point responses to the queries raised are presented below:

Major comments

1. My main concern is in relation to the genetic models the authors are working with in their study, as they are depleting their targets, IFITM3 or STAT1 in regulatory T cells from ontogeny, if these targets have an impact on the development or function of regulatory T cells it may generate results due to extrinsic cells that would mask the real

results. The authors should first analyse the phenotype (both percentage and number) of T populations in thymus, bone marrow, lymph nodes and spleen at the age of the studies and in the long term.

Response: Thanks for raising your concern. We have conducted the analysis of T cell percentage and phenotype in the thymus, spleen, bone marrow, and lymph nodes of the conditional knock-out mice used in our manuscript. The results showed that IFITM3 deficiency and STAT1 deficiency did not affect the development of T cells and the mice growth (Supplementary Figure 7a-e).

In suppl figure 2, how old are the animals under study? The author just showed the total percentage of CD4, CD8 or Treg but not their phenotype. The same study carry out on Spl Figure 3 has to be done in young mice in both KO models (IFITM3 and STAT1).

Another interesting option would be to work with inducible KO models or to generate mixed bone marrow chimeras, using a CD45.1/2 or CD90.1/2 reporter. Reconstitute C57BL/6 mice with 1:1 mixtures of congenically distinct CD45.1+ wild-type (WT) and CD45.2+ IFITM3 or STAT1 $-/-$ mature lineage-depleted (Lin $-$) bone marrow (BM) cells. (Roychoudhuri R et al, Nat Immunol, 2016. Doi : doi: 10.1038/ni.3441) be sure of the cell-intrinsic function of IFITM3/STAT1 in Treg.

Response: The mice used in Supplementary Figure 2 were 6-8 weeks-old male mice. We also analyzed the activation and cytokine secretion of T cells in WT and cKO

mice. The results have been added to the revised manuscript (Supplementary Figure 3a-d).

In addition, we agree with the reviewer that the phenotype study should also be carried out in STAT1 cKO mice, and the data have been shown in the revised manuscript (Supplementary Figure 7f-i). From the results, we concluded that STAT1 deficiency in Treg cells almost did not influence the development of Treg cells and the *in vivo* immune homeostasis.

2. To demonstrate the impact of IFN γ in the feedback loop of IFITM3 and STAT1, the authors work with a IFN γ ^{+/-} IFITM3^{fl/fl} animal and the results are shown in Figure 6e-1. As far as I understand the authors work with heterozygous for the non-conditional KO IFN γ mice, to confirm these results it would be necessary to repeat the same analysis but with a different technical approach.; such us anti-hIFN- γ neutralizing mAb.

Response: We appreciate the great suggestion from the reviewer. We used anti-mouse-IFN γ (BioXcell, BE0055) and the isotype control (BioXcell, BE0088) in the MC38 mice model, and the results were consistent with our previous findings in Figure 6 and Supplementary Figure. And we have added the results in the revised manuscript (Supplementary Figure 6i-k).

3.The authors should measure the antitumor specific response in their in vivo studies, regarding to KO IFITM3, KO STAT1, or double KO, Ex vivo stimulation of splenocytes against MC38 cell line, ELISPOT, ELISA should do to measure IFN γ or other cytokines. Parameters such as: number/mg tumor of tetramer positive (peptide MuLv,p15E KSPWF TTL), phenotype of T cell (CD44, CD62L, PD1, TIM3, CD137, ICOS) both tumor specific and unspecific should be measure within the tumour and periphery.

Response: Thanks for the comments. To mitigate the intolerable wait for tetramer procurement, we performed ELISA experiments to study the anti-tumor-specific response of IFITM3 cKO, SATT1 cKO, and double KO ex vivo stimulation of splenocytes against the MC38 cell line according to the system in the published paper (PMID:28885562). We measured the IFN γ level of the cell supernatant. The results have been added in the revised manuscript (Supplementary Figure 8a).

Minor comments

1. Figure 1, in MC38 murine model, comparison of peripheral Treg vs intratumor Treg, to demonstrated the role of IFIMT3 is depend on tumour microenvironment.

Response: Thanks for your comments. We have compared the phenotype of peripheral Treg and intratumor Treg, and we found no difference in peripheral Treg between WT and cKO mice; thus, we conclude that the role of IFIMT3 is mainly dependent on the tumor microenvironment (Fig3g-j).

2. Suppl 2d. The author should show the numbers of DP, DN, CD4, and CD8. Not only the percentage.

Response: Thanks for the comments. We have supplemented the cell numbers in Supplementary Figure 2d.

3. Figure 2, Boolean analysis has to be done to measure the T polyfunctionality instead a single positive analysis

Response: We appreciate the great suggestion from the reviewer. We have revised them in our manuscript (Figure 2g,h).

4. Suppl 3. At what month the naïve animals start to increase de effector T cell and IFN γ + cell?

Response: Thanks for your comments. We have detected the immune homeostasis of 8-week, 14-week, and 12-month-old mice. We did not find a difference in 8-week-old mice (Supplementary Figure 3c,d), but we found slightly higher IFN γ secretion in

14-week mice (Shown in the graph below). Therefore, we assume that the mice started to increase IFN γ ⁺ cells, and this change becomes progressively more pronounced with age.

5. In conditional KO models, is there any compensatory mechanism? How is the expression of IFITM10, IFITM2, IFITM1 on KO Treg cells?

Response: Thanks for the comments. We have tested the expression of the *Ifitm3* family in WT and KO Treg cells, and found that, except for *Ifitm3*, the expression of other genes did not show significant changes (Shown in the graph below). Thus, we assume that maybe there is no compensatory mechanism in KO models.

6. Figure 2, shown Survival curve

Response: Thanks for your comments. Tumor-induced lethality was defined as a tumor size reaching 225 mm² (PMID: 35143421) . Survival curves were analyzed by log-rank (Mantel-Cox) test. A *P* value of less than 0.05 was considered statistically significant. This figure is supplied as Figure 2c.

7. Test the suppression function of KO Treg in ex vivo experiments. Isolate de Treg from KO conditional mice (IFITM3 and STAT1) and coculture with effector T cells stimulate with CD3/CD28, analyses of cytokine production or proliferation rate of effector cell.

Response: Thanks for the comments. We isolated Treg cells and Tconv cells from WT, IKO, and SKO mice and performed an ex vivo suppression assay. From the results added in Supplementary Figure 8b, we found both IFITM3 deficient and STAT1 deficient in Treg cells could impair Treg suppression function, and double KO Treg restored the suppression function, which was consistent with our data from the mice tumor model. The same changes can be found in cytokine production (Supplementary Figure 8c,d).

8. Suppl Fig4, validation for RNASeq results should be done in intratumor Treg

Response: We agree with the reviewer, and performed RT-qPCR to validate the findings of RNASeq. The data representing the RT-qPCR analysis of intratumor Treg has been added in Figure 4c.

9. Some western blots throughout the article are not quantified.

Response: Thanks for the kind suggestion. We have quantified the western blot results and added the results to the manuscript.

10. Suppl Fig 5a, b. Demonstrate the interaction STAT1, IFITM3 with LC3B on Treg, not just in the HEK 293 cell line.

Response: Thanks for the kind comments. We have supplemented the interaction of STAT1, IFITM3 with LC3B in Treg as Supplementary Figure 5c.

11. Figure 5d.e. Demonstrate the interaction STAT1, JAK1 /JAK2 on Treg, not just in HEK 293 cell line

Response: Thanks for the kind suggestion. We have supplemented the interaction STAT1, JAK1 /JAK2 in Treg as Supplementary Figure 4e.

Reviewer #2 (Remarks to the Author):

IFITM3 has been shown to play a complex role in tumorigenesis of cancer cells. However,

the role and regulation of IFITM3 protein in tumor infiltrating regulatory T cells (TI-Tregs) is poorly understood.

The authors of this work have clearly characterized the effects of IFITM3-deficiency on Treg cells and described an IFN γ -dependent feedback loop governing the interactions of STAT1 with IFITM3. Crucially, IFITM3 conditional knockout Tregs were shown to lose their tumor-protective functions, such as IL10 or TGF β production, in favor of increased IFN γ expression, thus promoting anti-tumor responses.

Major points:

1. Although this paper focused on Tregs, demonstrating the positive regulation of IFITM3 by STAT1 has already been suggested in HCC cells (in <https://doi.org/10.1002/2211-5463.12479>). Furthermore, this same publication shows that IFITM3 can increase STAT1 levels in HCC cells, directly opposing what seems to be case in Tregs. The authors could address this and attempt to reconcile these contrasting observations.

Response: Thanks for your agreement with our work. We appreciate your comments. In this paper on IFITM3 function in HCC cells, IFITM3 regulates MMP9 expression through the p38/MAPK pathway and induces STAT1 expression in HCC cells, which was one of the downstream genes of the p38/MAPK pathway. p38/MAPK was implicated in the stress-induced phosphorylation of STAT1, whereas the IFN pathway was, in most cases, shown not to be dependent on p38 (PMID:10570180; 11226159). The paper published before (PMID: 12232043) showed that IFN γ stimulated phosphorylation of STAT1 occurs independently of the p38 pathway. IFN- γ -induced phosphorylation at Ser-727 as well as Tyr-701 was increased to similar levels in both p38 α (+/+) and p38 α (-/-) MEF cells. So, the IFITM3 deficiency in Treg cells induced higher phosphorylation of STAT1 independently of the p38 pathway and suppressed the autophagic degradation of STAT1, which may explain the different changes in STAT1 expression in HCC cells and Treg cells.

2. The graph in figure 5h could use a legend that indicates statistical significance. In addition, it could be worth exploring why pAKT (at the 473 serine) levels display higher values initially in cKO mice should the difference at t=0 be significant, as this is inconsistent with the published results showing IFITM3 is a PIP3 scaffold to amplify PI3K signaling

in B cells.

Response: Thank you for your comments. We have added a static legend in Figure 5h. Since we isolated the Treg cells from mice, the basal level of AKT phosphorylation in cKO Treg cells was already higher than WT Treg cells, so pAKT (at the 473 serine) levels display higher values initially in cKO mice at t=0.

In addition, the PI3K signaling is a downstream signal of the JAK-STAT1 signal (PMID: 30109213). In our data, IFITM3 could bind to the JAK-STAT protein complex to activate the PI3K-AKT pathway as well as decrease the FOXO1 protein level. So the effect of IFITM3 on JAK-STAT1 was stronger than the effect of IFITM3 on PI3K-AKT pathway, which may explain the inconsistent changes of PI3K-AKT signaling pathway in Treg cells and B cells.

3. Whether STAT1 affects Treg stability (Foxp3, Ifng expression or suppressive assay) should be determined. In addition, Treg cell stability needs also to be checked in DKO mice.

Response: Thanks for your constructive comments. We first analyzed the Treg stability and T cell homeostasis in WT, SKO, and DKO mice (Supplementary Figure 7). We also displayed the suppression assay of Treg cells from STAT1 KO mice and DKO mice, and the data has been shown in the revised manuscript (Supplementary Figure 8b-d). From the results, we concluded that STAT1 deficiency in Treg cells almost did not influence the development of Treg cells and the *in vivo* immune homeostasis.

4. The expression of IFITM3 in CD4⁺ effective T cells and CD8⁺ CTLs in the tumor should also be determined. The potential functions of IFITM3 in these cells should be discussed.

Response: Thanks for your kind suggestion. We are equally curious about the function of IFITM3 in tumor-infiltrating CD4⁺ T cells and CD8⁺ T cells. We performed the RT-qPCR analysis of *Ifitm3* expression in CD4⁺ and CD8⁺ effector cells, and found that the relative expression of *Ifitm3* was higher in Teff cells and Treg (Supplementary Figure 9a). Then, we hypothesized that IFITM3 may also play a vital role in CD4⁺ and CD8⁺ T cells. Therefore, we crossed the *Ifitm3*^{fl/fl} mice with *Cd4*^{cre} mice to address the function of IFITM3 in CD4⁺ and CD8⁺ T cells. IFITM3 deficiency in CD4⁺ T cells and CD8⁺ T cells secret higher levels of IFN γ after TCR stimulation (Supplementary Figure 9b,c). The proliferation and the apoptosis results indicated that the IFITM3 deficient T cells were more proliferative (Supplementary Figure 9d). In addition, we found the naïve CD4⁺ T cells from KO mice preferred to differentiate into Th1 cells but not Treg cells, which was consistent with our findings in Treg conditional knock-out mice (Supplementary Figure 9e,f).

We also inoculated WT and KO mice with MC38 murine colon cancer cells subcutaneously to determine the function of IFITM3 *in vivo*. The KO mice exhibited a profound reduction in MC38 tumor size and showed increased secretion of IFN γ ⁺ cytokines (Supplementary Figure 9g-j).

Minor points:

1. In the last paragraph of the introduction, when referring to IFITM3 levels, “deregulated” could be substituted for another term that better denotes a deficiency as it can be

interpreted as an overexpression.

Response: Thank you for your suggestions. We have changed “deregulated” to “reduced”.

2. Why were no co-immunoprecipitation experiments done to confirm the interaction of IFITM3 with JAK1 and JAK2 as it was the case for STAT1?

Response: Thanks for the kind suggestion. We have supplemented the interaction of IFITM3 with STAT1, JAK1 /JAK2 in Treg. The data is shown in Supplementary Figure 4e.

Reviewer #3 (Remarks to the Author):

This manuscript presents a very important study with a large amount of data that largely support the main conclusions. This work carries substantial impact on our understanding of the complexity of the immunosuppressive function and fragility of Treg cells in the tumor microenvironment. However, there are a few minor concerns that need to be addressed to improve the clarity and rigor of this work.

Response: Thanks for your agreement with our work and for raising minor concerns. We have revised the manuscript according to your kind suggestions.

1. What are specifically the matched normal tissues used in comparison to tumor tissues in Fig 1? Are they isolated from the same tumor patients or healthy control subjects?

Response: The normal tissue in our experiments was the normal tissue from the same tumor patients. Normal tissue adjacent to the tumors was collected from the same patients (5 cm at least away from the tumor edge).

2. It appeared that the in vivo tumor growth results shown in Fig 6e-f and Fig 7a-b were performed only once with small number of mice per group (n=4 or 5). As these results are essential for the respective conclusions, replicate experiments should be performed to confirm the phenotypes.

Response: Thanks for your comments. We have added the experimental data in Figure 6e and Figure 7a to confirm the tumor phenotypes (n=7).

3. There are many sentences in the manuscript that are problematic in grammar or not accurate scientifically. For example, the first sentence in the results section, “Tumor-infiltrating Treg cells feature correlation expression of IFITM3 and FOXP3”.....the first sentence in the last paragraph of discussion section, “In summary, we identified the feedback loop of STAT1-IFITM3 maintained Treg function and stability”. The first authors and correspondence authors should carefully go over the whole manuscript to make all necessary corrections.

Response: Thanks for your comments. We have carefully reviewed and revised the entire manuscript to improve the grammar and overall readability.

REVIEWERS' COMMENTS

Reviewer #1 (Remarks to the Author):

Thank you to the authors for their thorough review of the article. The authors have responded to all of my comments.

I agree with its publication.

Reviewer #2 (Remarks to the Author):

The authors have invested considerable effort in addressing the research questions, resulting in significant improvements to the manuscript. No additional questions have arisen, and it is now suitable for publication without further delay.

Reviewer #3 (Remarks to the Author):

The authors have addressed all comments and the revised version is largely improved.

REVIEWERS' COMMENTS

Reviewer #1 (Remarks to the Author):

Thank you to the authors for their thorough review of the article. The authors have responded to all of my comments.

I agree with its publication.

Response: Thank you for your approval and your help with our manuscript.

Reviewer #2 (Remarks to the Author):

The authors have invested considerable effort in addressing the research questions, resulting in significant improvements to the manuscript. No additional questions have arisen, and it is now suitable for publication without further delay.

Response: Thank you for your approval and your help with our manuscript.

Reviewer #3 (Remarks to the Author):

The authors have addressed all comments and the revised version is largely improved.

Response: Thank you for your approval and your help with our manuscript.